# Dynamics of macrophage polarization support *Salmonella* persistence in a whole living organism

Jade Leiba[1]*, Tamara Sipka[1†], Christina Begon-Pescia[1†], Matteo Bernardello[2], Sofiane Tairi[1], Lionello Bossi[3], Anne-Alicia Gonzalez[4], Xavier Mialhe[4], Emilio J Gualda[2], Pablo Loza-Alvarez[2], Anne Blanc-Potard[1], Georges Lutfalla[1], Mai E Nguyen-Chi[1]*

[1]LPHI, Université de Montpellier, CNRS, INSERM, Montpellier, France; [2]ICFO - Institute of Photonic Sciences, The Barcelona Institute of Science and Technology, Castelldefels, Barcelona, Spain; [3]Institute for Integrative Biology of the Cell-I2BC, Université Paris-Saclay, CEA, CNRS, Gif-sur-Yvette, France; [4]MGX-Montpellier GenomiX, Université de Montpellier, CNRS, INSERM, Montpellier, France

**\*For correspondence:**
jade.leiba@orange.fr (JL);
mai-eva.nguyen-chi@
umontpellier.fr (MEN-C)

[†]These authors contributed equally to this work

**Competing interest:** The authors declare that no competing interests exist.

**Abstract** Numerous intracellular bacterial pathogens interfere with macrophage function, including macrophage polarization, to establish a niche and persist. However, the spatiotemporal dynamics of macrophage polarization during infection within host remain to be investigated. Here, we implement a model of persistent *Salmonella* Typhimurium infection in zebrafish, which allows visualization of polarized macrophages and bacteria in real time at high resolution. While macrophages polarize toward M1-like phenotype to control early infection, during later stages, *Salmonella* persists inside non-inflammatory clustered macrophages. Transcriptomic profiling of macrophages showed a highly dynamic signature during infection characterized by a switch from pro-inflammatory to anti-inflammatory/pro-regenerative status and revealed a shift in adhesion program. In agreement with this specific adhesion signature, macrophage trajectory tracking identifies motionless macrophages as a permissive niche for persistent *Salmonella*. Our results demonstrate that zebrafish model provides a unique platform to explore, in a whole organism, the versatile nature of macrophage functional programs during bacterial acute and persistent infections.

## Editor's evaluation

This useful study introduces the development of *Salmonella* infection model in zebrafish embryos as an important model to study the interaction between macrophages and *Salmonella* during in vivo infection. Overall, the data presented are convincing and provide an inventory of genes mediating macrophage cell-cell adhesion and interactions that are useful for dissecting tissue macrophage responses and heterogeneity during intracellular bacterial infection. This is important to characterise the infection outcome and the dynamics of the immune response. The work will be of interest to microbiologists.

## Introduction

The outcome of bacterial infections is the result of complex dynamic interactions between the pathogen and the host's cellular and humoral actors of innate immunity. Deciphering this complexity is necessary to predict infection outcomes and guide therapeutic strategies. The development of

tractable systems in which bacteria and cellular actors can be tracked at high spatiotemporal resolution in a whole living animal is essential to assess the dynamic of host–pathogen interactions.

*Salmonella enterica*, a Gram-negative facultative intracellular pathogen, comprises more than 2000 non-typhoidal and typhoidal serovars inducing a variety of conditions ranging from benign gastroenteritis to severe systemic infection (*Gogoi et al., 2019*). Every year typhoidal serovars infect 20 millions of people and cause more than 200,000 of deaths (*Crump et al., 2004*). In some cases, *Salmonella* establishes a chronic infection that results in asymptomatic carriers hosting the environmental reservoir for further infections (*Monack, 2012*; *Ruby et al., 2012*; *Gal-Mor, 2019*). *Salmonella* resides within the draining lymph nodes and systemic tissues such as the spleen, where the bacteria survive mainly inside macrophages and have to deal with this hostile environment (*Eisele et al., 2013*; *Garai et al., 2012*; *Ehrhardt et al., 2023*). Capitalizing on different virulence factors, *Salmonella* can replicate within macrophages inside modified vacuoles called phagosomes and escape the host's defenses (*LaRock et al., 2015*). In addition, as reported for several intracellular bacterial pathogens, *Salmonella* uses strategies to interfere with macrophage polarization (*Thiriot et al., 2020*).

Macrophages are among the most plastic immune cells that adapt their phenotype and function according to their microenvironment by a process called polarization. In vivo, they form a continuum of activation states whose two extremes are the pro-inflammatory M1 macrophages, that have a bactericidal activity, and the anti-inflammatory M2 macrophages that promote the resolution of inflammation and healing (*Ginhoux et al., 2016*). During the first hours of infection, most bacteria, including *Salmonella enterica* serovar Typhimurium (*S*. Typhimurium), induce macrophage polarization toward the pro-inflammatory and microbicidal M1 phenotype (*Jenner and Young, 2005*; *Nau et al., 2002*). In contrast, persisting bacteria, like *Mycobacterium tuberculosis*, *Brucella abortus*, or *S*. Typhimurium, may preferentially reside inside healing/anti-inflammatory M2 macrophages (*Eisele et al., 2013*; *Thiriot et al., 2020*; *Xavier et al., 2013*). In a mouse model of *S*. Typhimurium long-term infection, bacteria were shown to persist mainly in M2 macrophages (*Eisele et al., 2013*; *Pham et al., 2020*). In contrast, another in vivo study showed that *S*. Typhimurium reside in inducible nitric oxide synthase (iNOS)-expressing macrophages that clustered within splenic granulomas 42 days post-infection in mice (*Goldberg et al., 2018*). Although iNOS is a known marker of M1 macrophages, the exact polarization status of these macrophages remained undetermined. In addition, intracellular replication of *S*. Typhimurium was shown to vary according to macrophage polarization, with a greater replication in M2 macrophages (*Saliba et al., 2016*). To date, however, the dynamics of interactions between pathogenic bacteria and polarized macrophages between early invasion and late survival remain poorly understood at the organism level.

Perfectly suited to live observation of immune cells and pathogens in vivo, the transparent zebrafish embryo has emerged as a powerful vertebrate model to study host–pathogen interactions at the cellular and whole organism level (*Torraca and Mostowy, 2018*). This model allows macrophage plasticity and reprogramming to be monitored using dedicated fluorescent reporters to simultaneously tract macrophages and visualize their activation state. Using a non-infected wound model, we previously showed that macrophages first express pro-inflammatory cytokines (M1-like polarization) before switching to a new state expressing both M1 and M2 makers during the wound healing process (*Nguyen-Chi et al., 2015*). Yet, the dynamics of macrophage polarization in the context of *S*. Typhimurium infection has not been addressed in zebrafish.

Here, we establish the first larval zebrafish model of *S*. Typhimurium persistent infection. Using intravital imaging, we show that during early stages of infection, both neutrophils and macrophages are mobilized to the infection site to engulf bacteria and that macrophages respond by a strong M1-like activation. In later stages of infection, bacteria survive inside non-inflammatory macrophages which accumulate in large clusters and provide a niche for persistent bacteria inside the host. Finally, a comprehensive analysis of the transcriptional profiles of macrophages reveals a highly dynamic transcriptional signature of distinct macrophage subsets during early and late phases of infection, showing changes in inflammatory response and in expression of adhesion molecules upon persistent infection, which correlate with a decrease of macrophage motility. This new model provides a unique opportunity to explore the dynamics of interactions between persistent pathogenic bacteria and polarized macrophages in a four-dimentional living system.

## Results

### *Salmonella* hindbrain ventricle infection in zebrafish leads to different outcomes, from systemic to persistent infection

Previous studies on *S*. Typhimurium-infected zebrafish larva, based on intravenous or subcutaneous injection or immersion, have been linked to a rapid disease progression with acute symptoms and high production of pro-inflammatory cytokines, leading to larval mortality (*Stockhammer et al., 2009*; *van der Sar et al., 2003*). To develop a model of persistent bacterial infection using *S*. Typhimurium (hereafter named *Salmonella*), we chose to inject in a closed compartment, the hindbrain ventricle (HBV) (*Figure 1A*), which allows direct visualization of immune cell recruitment in a confined space. To allow direct observation of bacterial burden, a GFP-expressing *Salmonella* ATCC14028s strain was constructed (Sal-GFP). We injected 2 days post-fertilization (dpf) zebrafish embryos with different doses of Sal-GFP in the HBV, or with phosphate buffered saline (PBS) as control (*Figure 1B*) and monitored larval survival from 0 to 4 days post-infection (dpi). Increasing injection doses of *Salmonella* resulted in increased larval mortality. When less than 500 colony-forming units (CFU) of Sal-GFP were injected, all zebrafish larvae survived the infection (*Figure 1C*). On the other hand, HBV injection of 1000–2000 CFU of Sal-GFP led to 50% of larvae survival at 4 dpi (*Figure 1D*).

To evaluate the bacterial burden in infected hosts, the number of CFU was counted every day from 0 to 4 dpi after HBV injection of Sal-GFP (*Figure 1E*). Before processing for CFU counting, infected larvae and their controls were imaged individually by fluorescence microscopy. No CFU were detected in the PBS-injected larvae (*Figure 1—figure supplement 1A*). After injecting 1000–2000 CFU, bacteria efficiently proliferated in 47% of infected larvae, from 1 to 4 dpi (50,000–200,000 CFU, *Figure 1E, G*). Those larvae developed a systemic infection with *Salmonella* proliferation in the HBV and other tissues, including notochord, heart, and circulating blood (*Figure 1F*) and usually died between 2 and 4 dpi. Among the 53% of infected larvae that survived the infection at 4 dpi, 40% harbored live bacteria (10–2000 CFU), while 36% were bacteria free (*Figure 1E, G*). When injecting less than 500 CFU, no larva exhibited bacterial hyper-proliferation (*Figure 1—figure supplement 1B*) and 75% showed evidence of *Salmonella* persistence at 4 dpi (10–1000 CFU, *Figure 1—figure supplement 1B*). Importantly, alive infected larvae still harbored persistent bacteria at 14 dpi (*Figure 1—figure supplement 1C*).

Altogether, these results show that *Salmonella* infection in HBV leads to different outcomes, that we classified as: (1) High Proliferation, indicating a high bacterial burden with systemic infection leading to death; (2) Infected, indicating surviving larvae with persisting bacteria; and (iii) Cleared, indicating that larvae completely recovered with no more detectable bacteria (*Figure 1G*).

Throughout the remainder of this study, an optimized infection dose of 1000–1500 CFU was injected into the HBV of 2 dpf embryos and we further focused (unless otherwise stated) on the so-called 'infected' cohort with established persistent infection.

### The global host inflammatory response to *Salmonella* HBV infection

To investigate the global host immune response to *Salmonella* infection in the zebrafish model, the relative expression of several immune-related genes was examined by qRT-PCR within whole larvae from 3 hpi to 4 dpi after *Salmonella* challenge (*Figure 2* and *Figure 2—figure supplement 1*). At 3 hpi, infected larvae showed elevated levels of expression of pro-inflammatory cytokines: *interleukin-1 beta* (*il-1b*), *tumor necrosis factor a and b* (*tnfa* and *tnfb*, two orthologs of mammalian *TNF*), *interleukin-8* (*il-8*), and of the inflammation marker *matrix metalloproteinase 9* (*mmp9*) (*Figure 2A*), consistent with previous findings in zebrafish models of systemic *Salmonella* infection (*Stockhammer et al., 2009*). Subsequently, pro-inflammatory gene expression (*il-1b*, *tnfb*, *il-8*, and *mmp9*) significantly decreased at 1 dpi and raised from 2 hpi to 4 dpi to reach similar levels to those detected at 3 hpi. At 4 dpi, *tnfa* expression was still up-regulated in infected larvae compared to PBS controls (*Figure 2—figure supplement 1*). *Salmonella* infection induced *ccl38a.4* gene, encoding CCL2, a key chemokine of macrophage migration that binds the CCR2 receptor (*Cambier et al., 2014*) at 4 hpi but has no effect on *cxcr4b* and *sdf1* (*cxcl12a*), suggesting a role of the Ccl2/Ccr2 axis in macrophage mobilization to the infection site (*Figure 2A* and *Figure 2—figure supplement 1*). The up-regulation of the macrophage-specific marker *mfap4* during the time course of infection highlighted an overall macrophage response (*Figure 2A*). In contrast, we found that the other frequently used macrophage-specific marker *mpeg1* was down-regulated early after *Salmonella* infection (*Figure 2A*), as previously

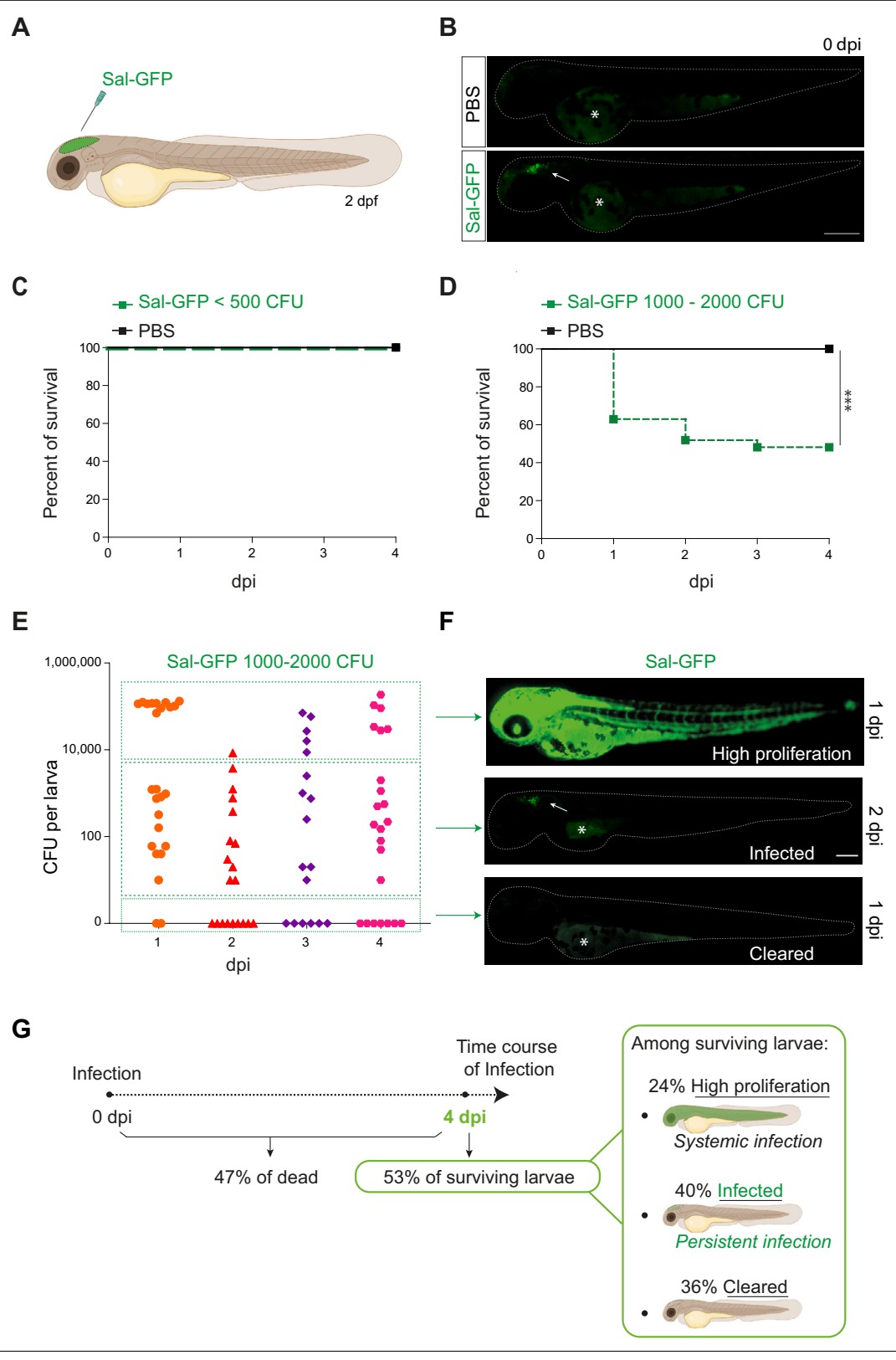

**Figure 1.** Zebrafish is a pertinent model for persistent *Salmonella* infection. (**A**) Schematic illustration of 2 dpf zebrafish embryo infected in the hindbrain ventricle (HBV) with Sal-GFP, a GFP-expressing strain. (**B**) Representative fluorescent images of HBV-injected larvae with either PBS or 1000 CFU of Sal-GFP shortly after microinjection. White arrow: bacteria in the HBV. Dots outline the larva. Asterisk: auto-fluorescence of the yolk. Scale bar: 200 μm.

*Figure 1 continued on next page*

*Figure 1 continued*

Survival curves of injected embryos with either PBS or different doses of Sal-GFP, that is (**C**) <500 CFU or (**D**) 1000–2000 CFU. One representative of three replicates (*n* = 24 larvae per condition). Log rank test, ***p < 0.001. (**E**) CFU counts per embryos infected with a range of 1000–2000 CFU of Sal-GFP at 1, 2, 3, and 4 dpi. Pool of four independent experiments ($n_{1\,dpi}$ = 25, $n_{2\,dpi}$ = 20, $n_{3\,dpi}$ = 20, $n_{4\,dpi}$ = 25 larvae). Kruskal–Wallis test (unpaired, non-parametric): not significant. (**F**) Representative fluorescent images of Sal-GFP-infected larvae. Bacteria are in green. Dots outline the larva. Asterisk: auto-fluorescence of the yolk. Scale bar: 200 µm. (**G**) Schematic representation of the different infection outcomes, High Proliferation, Infected, and Cleared, induced by injection of 1000–2000 CFU of Sal-GFP. From 0 to 4 dpi, 47% of the infected larvae developed a systemic infection where the bacteria displayed highly proliferation leading to larval death (High Proliferation). At 4 dpi, among the surviving larvae, 24% still exhibited a systemic infection, while 36% recovered from the infection with no detectable CFU (Cleared) and 40% contained persistent bacteria (Infected).

The online version of this article includes the following figure supplement(s) for figure 1:

**Figure supplement 1.** *Salmonella* localized infection leads to persistence in zebrafish embryos.

---

described during *Salmonella* and *Mycobacterium marinum* infections in zebrafish (*Benard et al., 2015*). Then, the kinetics of expression of anti-inflammatory genes revealed a down-regulation of *mannose receptor, C type 1b* (*mrc1b*) early after *Salmonella* infection. Intriguingly, relative levels of *mrc1b* expression increased from 3 hpi to 4 dpi. Furthermore, *Negative Regulator of Reactive Oxygen Species* (referred to as *nrros*), regulator of reactive oxygen species and of TGF-b in mammals (*Liu et al., 2013*; *Ma et al., 2019*; *Noubade et al., 2014*; *Qin et al., 2018*), showed kinetic of expression similar to *mrc1b* during *Salmonella* infection.

Taken together these results highlight a specific immune response triggered by *Salmonella*, and define two main phases: an early pro-inflammatory phase characterized by the induction of pro-inflammatory markers and down-regulation of anti-inflammatory markers, and a late phase (between 3 and 4 dpi) characterized by de novo up-regulation of pro-inflammatory gene expression concomitant with the increased expression of anti-inflammatory genes (*Figure 2B*). Based on these results, we decided to further focus our study on deciphering the immune response to *Salmonella* infection both during the early phase, few hours post-infection, and during the late phase at 4 dpi.

## Early phase of *Salmonella* HBV infection induces strong macrophage and neutrophil responses

Among immune cells, macrophages are a niche for *Salmonella* infection in mammals (*Gogoi et al., 2019*; *Garai et al., 2012*) and in zebrafish (*Stockhammer et al., 2009*; *van der Sar et al., 2003*). To investigate the role of macrophages in the control of *Salmonella* infection, macrophage reporter embryos, Tg(*mfap4:mCherry-F*) were injected in the HBV with Sal-GFP or PBS and the global macrophage population was imaged by fluorescence microscopy from 3 hpi to 4 dpi (*Figure 3A*). We noticed that *Salmonella* infection led to an increase in the global macrophage population that persisted over days compared to PBS-injected larvae, suggesting that *Salmonella* infection triggers myelopoiesis (*Figure 3A, B*). At 3 hpi macrophages were specifically recruited to the HBV of infected embryos. In addition, at 4 dpi, an intense macrophage accumulation persisted in the brain of infected embryos, in line with the development of a *Salmonella* persistent localized infection. To test whether *Salmonella* persistence depends on the infection site, we injected either Sal-GFP or PBS in the otic vesicle of 2 dpf macrophage reporter embryos, Tg(*mfap4:mCherry-F*) (*Figure 3—figure supplement 1A*). At 4 dpi, intravital confocal imaging revealed macrophage recruitment to the infected otic vesicle and the presence of bacteria, confirming the intrinsic capacity of *Salmonella* to persist within the host for a long time (*Figure 3—figure supplement 1B*).

To obtain a 4D (space + time) description of macrophage interactions with *Salmonella*, we used a light sheet fluorescence microscopy (LSFM) system, convenient to study the 3D architecture of cells over a large field of view and long periods of time with minimal side effects. Tg(*mfap4:mCherry-F*) larvae were infected with Sal-GFP and imaged. The resulting video sequences revealed the macrophage recruitment within 2 hpi (*Figure 3C* and *Video 1*) and immediate *Salmonella* internalization (*Figure 3C*, arrowheads, *Figure 3D* and *Video 2*).

To investigate the role of macrophages in the control of *Salmonella* infection, we ablated macrophages using Tg(*mpeg1:Gal4/UAS:nfsB-mCherry*) embryos and metronidazole (MTZ) treatment

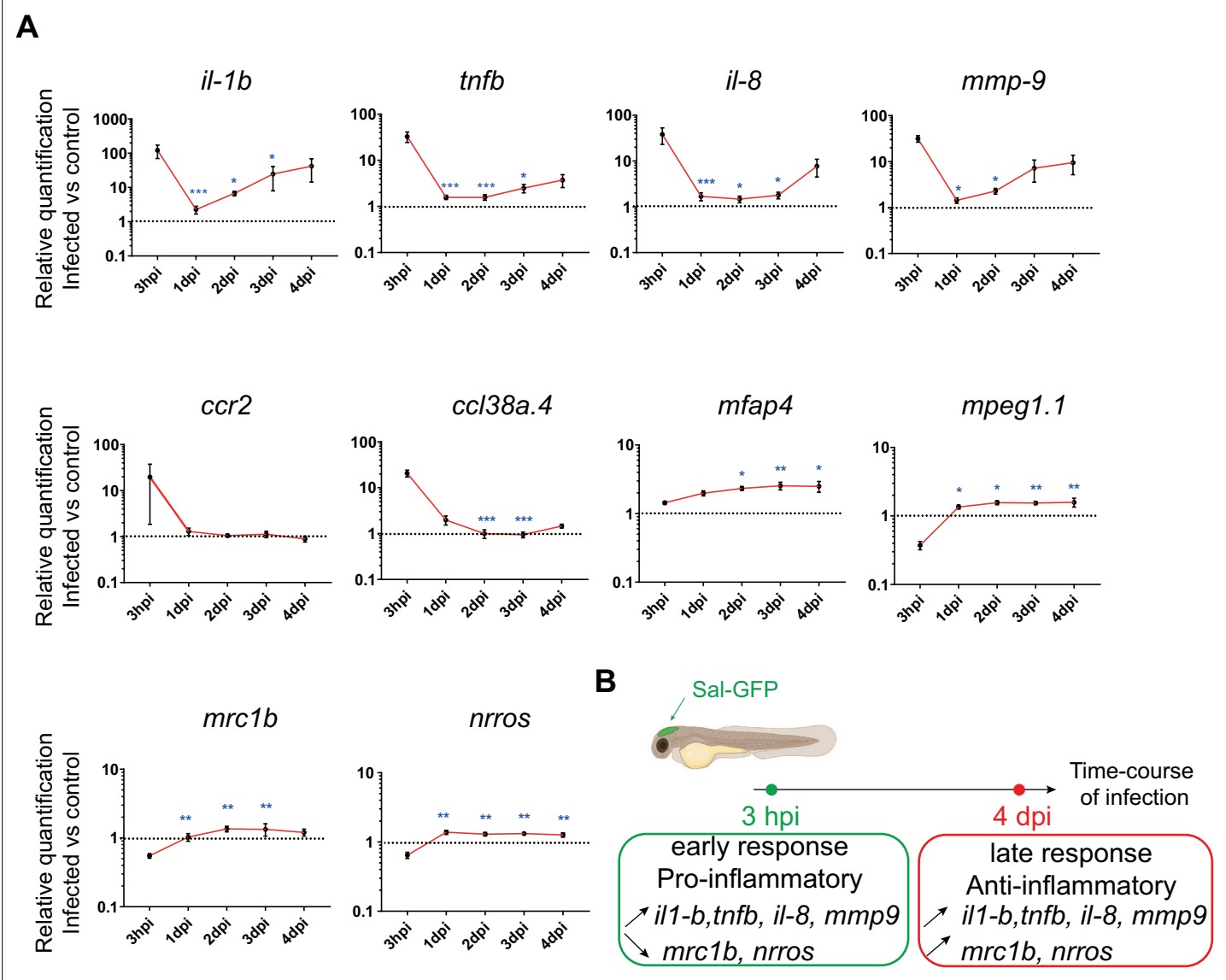

**Figure 2.** The global host inflammatory response to *Salmonella* infection. (**A**) RT-qPCR analysis of *il1b*, *tnfb*, *il8*, *mmp9*, *ccr2*, *ccl38a.4*, *mpeg1*, *mfap4*, *mrc1b*, and *nrros*, mRNA expression infected versus non-infected, normalized with *ef1a*. Larvae were either PBS- or Sal-GFP injected and RNA samples were extracted from whole larvae at 3 hpi, 1, 2, 3 and 4 dpi. After infection, larvae displaying 'high proliferation' of bacteria or bacteria 'cleared' were excluded from the analysis. Data are presented as relative expression in the infected larvae compared with the relevant PBS-injected controls ($2^{-\Delta\Delta C_P}$). Values are the means ± standard error of the mean (SEM) of eight replicates (*n* = 8 larvae per time point). Kruskal–Wallis test (unpaired, non-parametric). *p < 0.05; **p < 0.01; ***p < 0.001 show significant differences compared to 3 hpi. (**B**) Diagram of global host inflammatory response to *Salmonella* infection.

The online version of this article includes the following figure supplement(s) for figure 2:

**Figure supplement 1.** Expression of *tnfa*, *cxcr4b*, *cxcl12a* mRNA during *Salmonella* infection.

(*Nguyen-Chi et al., 2020*; *Figure 3—figure supplement 2A, B*). MTZ (added at 48 hr post-fertilization (hpf)) specifically ablated 71% of macrophages in transgenic larvae after 24 hr of treatment (*Figure 3— figure supplement 2, D*). DMSO treatment on transgenic larvae (NTR+MTZ−) and MTZ treatment on WT embryos (NTR−MTZ+) were used as controls (*Figure 3—figure supplement 2C–H* and not shown). PBS injection had no effect on the mortality of embryos from the different groups, confirming the absence of toxicity of MTZ treatment (*Figure 3—figure supplement 2E*). After infection with a sublethal dose of 500 CFU of Sal-GFP, MTZ-mediated macrophage depletion increased larval mortality up to 54% at 4 dpi while control group showed 4% mortality (*Figure 3—figure supplement*

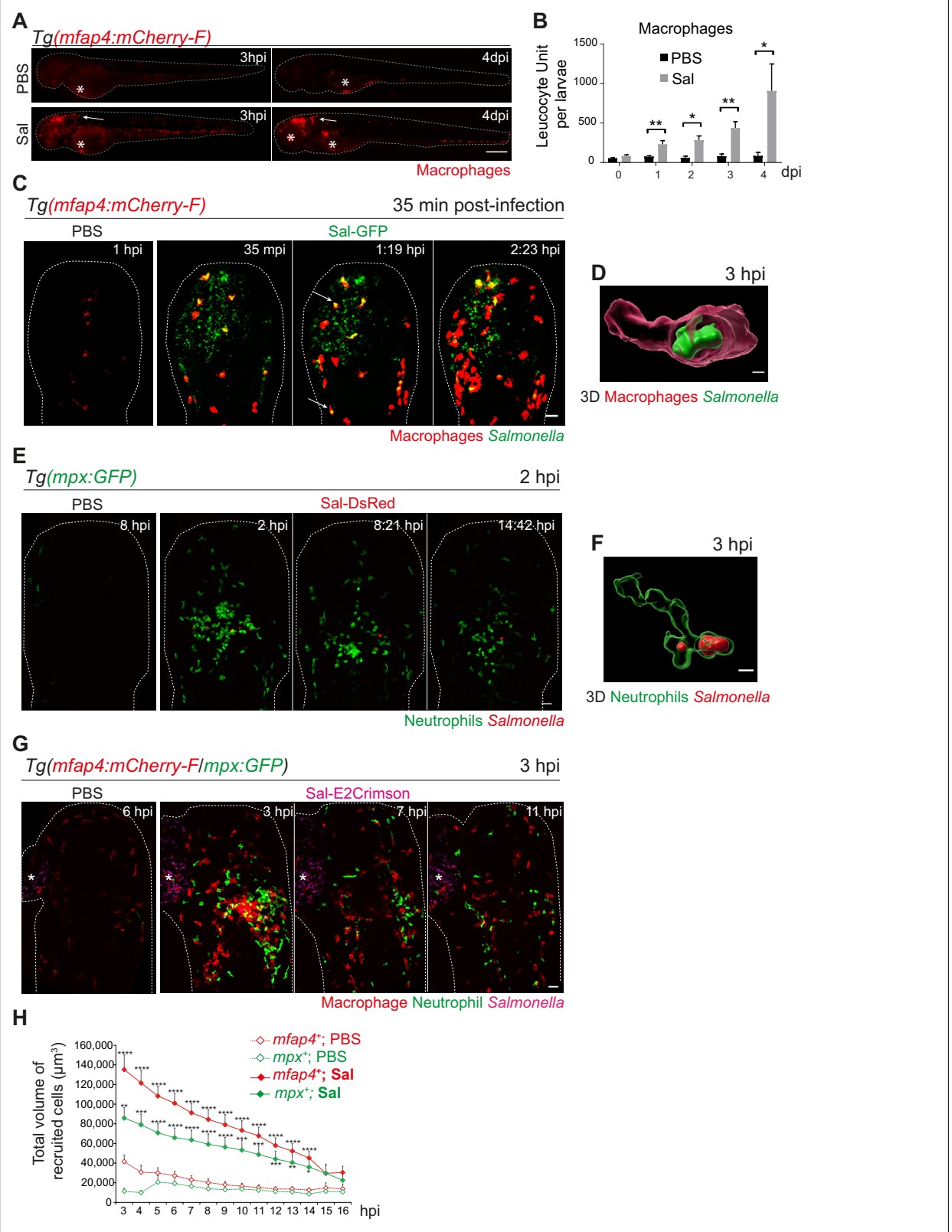

**Figure 3.** Early phase of *Salmonella* hindbrain ventricle (HBV) infection induces strong macrophage and neutrophil responses. (**A–D**) Tg(*mfap4:mCherry-F*) larvae were injected with either PBS or Sal-GFP in HBV. (**A**) Representative fluorescent images of larvae showing macrophage recruitment at the site of injection at 3 hpi and at 4 dpi. Asterisk: auto-fluorescence. Scale bar: 200 µm. (**B**) Quantification of total macrophages at 0, 1, 2, 3, and 4 dpi. One representative of three replicates (mean number of leukocyte units/larva ± SEM, $n_{0\,dpi}$ = 24, $n_{1\,dpi}$ = 11, $n_{2\,dpi}$ = 6, $n_{3,4\,dpi}$ = 5 per condition,

*Figure 3 continued on next page*

*Figure 3 continued*

Mann–Whitney test, two-tailed, *p < 0.05, **p < 0.01). (**C**) Representative maximum projections of fluorescent images extracted from 4D sequences using light sheet fluorescence microscopy starting 35 min post-infection during 2 hr, showing recruitment of macrophages (red) to the infection site (*Salmonella*, green). Scale bar: 30 µm. (**D**) 3D reconstruction of a macrophage phagocytosing *Salmonella* at 3 hpi. Scale bar: 5 µm. (**E, F**) Tg(*mpx:GFP*) larvae were injected with PBS or Sal-DsRed in HBV. (**E**) Representative maximum projections of fluorescent images extracted from 4D sequences using confocal microscopy at 2 hpi during 13 hr, showing recruitment of neutrophils (green) to the infection site (*Salmonella*, red). Scale bar: 35 µm. (**F**) 3D reconstruction of a neutrophil phagocytosing *Salmonella* at 2 hpi. Scale bar: 5 µm. (**G, H**) Tg(*mfap4:mCherry-F/mpx:GFP*) larvae were injected with either PBS or Sal-E2Crimson in HBV. (**G**) Representative maximum projections extracted from 4D sequences using confocal microscopy from 3 to 14 hpi showing recruitment of both neutrophils (green) and macrophages (red) to the infection sites. Asterisk: auto-fluorescence. Scale bar: 50 µm. (**H**) Quantification of the total volume of recruited cells (*mfap4*+ or *mpx*+ cells) from 3 to 16 hpi. Data of three replicates pooled (mean volume/larva ± SEM, n = 11 from 3 to 4 hpi, n = 15 from 5 to 14 hpi, n = 4 from 15 to 16 hpi per condition, Mann–Whitney test, two-tailed, significance of Sal versus PBS conditions *p < 0.05, **p < 0.01, ***p < 0.001, ****p < 0.0001).

The online version of this article includes the following figure supplement(s) for figure 3:

**Figure supplement 1.** Macrophage response upon *Salmonella* injection in the otic vesicle.

**Figure supplement 2.** Macrophages are essential to control *Salmonella* infection in zebrafish.

*2F*). MTZ-mediated macrophage depletion also strongly impacted bacterial clearance with development of a systemic infection, as shown by fluorescence microscopy and quantification of bacterial burden (*Figure 3—figure supplement 2G, H*). Altogether, these data show that macrophages are instrumental to control *Salmonella* infection in HBV.

Neutrophils are also innate immune players in the defense against *Salmonella* in various models (*Cheminay et al., 2004*; *Hall et al., 2012*). To investigate their role in our model, we infected Tg(*mpx:GFP*) neutrophil reporter embryos with DsRed-fluorescent *Salmonella* (Sal-DsRed). The fluorescence analysis of global neutrophil populations showed an increase within 2 dpi, in line with the reported *Salmonella*-induced granulopoiesis (*Hall et al., 2012*) (data not shown). Time-lapse imaging was used to visualize early neutrophil–bacteria interactions (*Figure 3E* and *Video 3*): while neutrophils were absent from PBS-injected HBV, they were immediately recruited in the

**Video 1.** *Salmonella* early-infection induces a strong macrophage response. Tg(*mfap4:mCherry-F*) larvae were infected with Sal-GFP in hindbrain ventricle. Time-lapse videos of labeled macrophages were acquired using light sheet microscopy at 35 min post-infection during 2 hr and image series were collected every 1 min. One representative movie (maximum projections) is presented, showing the recruitment of macrophages (*mfap4*+ cells, red) to the infection site (*Salmonella*, green). Time is indicated in the top left corner. Scale bar: 30 µm.

https://elifesciences.org/articles/89828/figures#video1

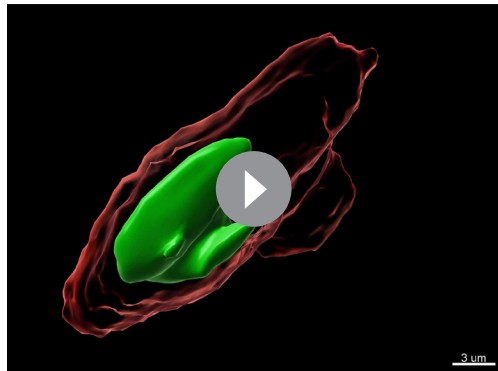

**Video 2.** Macrophages are able to phagocytose *Salmonella* during early stages of infection. Tg(*mfap4:mCherry-F*) larvae were infected with Sal-GFP-int in hindbrain ventricle and imaged by confocal microscopy at 3 hpi. This movie is a 3D reconstruction animation created from representative fluorescent confocal images using Imaris software and shows a single macrophage (red) that have phagocytosed *Salmonella* (green) at 3 hpi. Scale bar: 5 µm.

https://elifesciences.org/articles/89828/figures#video2

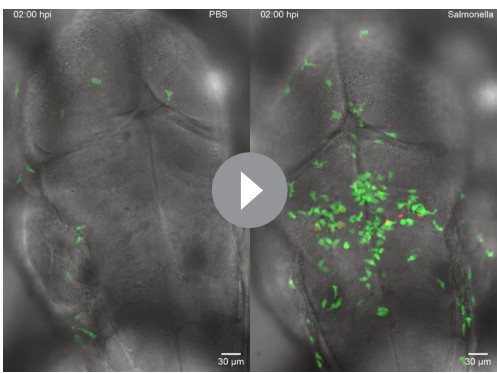

**Video 3.** *Salmonella* early infection induces a strong neutrophil response. Tg(*mpx:GFP*) larvae were injected with PBS or Sal-DsRed in hindbrain ventricle. Time-lapse videos of labeled neutrophils were acquired by confocal microscopy at 2 hpi during 13 hr and image series were collected every 3 min. Two representative movies (maximum projections) of PBS-injected larva (left panel) and *Salmonella*-infected larva (right panel) are presented, showing the recruitment of neutrophils (*mpx*⁺ cells, green) to the infection site (*Salmonella*, red). Time is indicated in the top left corner of each panel. Scale bar: 30 µm.

https://elifesciences.org/articles/89828/figures#video3

infected HBV and participated to bacterial clearance through phagocytosis. Subsequently, after 3 hpi, the number of neutrophils decreased at the infection site (*Figure 3E, F*).

To assess the relative contribution of both macrophages and neutrophils to control *Salmonella* infection, Tg(*mfap4:mCherry-F/mpx:GFP*) embryos were infected in the HBV with E2Crimson fluorescent *Salmonella* (Sal-E2Crimson) and imaged from 3 to 14 hpi (*Figure 3G* and *Video 4*). Because counting individual leukocytes that stack together at the infection site is not reliable, we measured the total of volume of both macrophages (*mfap4*⁺ cells) and neutrophils (*mpx*⁺ cells) using 3D reconstructions. Total volume analysis showed that macrophages and neutrophils were both strongly recruited to the infection site in the first hours of infection compared to controls and that their mobilization slightly diminished over time post-infection with similar kinetics (*Figure 3G, H*).

Altogether, we concluded that both macrophages and neutrophils actively participate to the immediate innate immune response to *Salmonella* infection in the HBV, characterized by myelopoiesis, granulopoiesis, leukocyte recruitment to the infection site, and phagocytosis of bacteria.

## *Salmonella* HBV infection induces hyper-accumulation of macrophages harboring persistent bacteria at late stage of infection

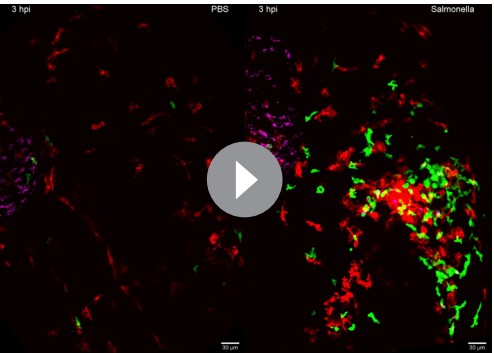

**Video 4.** Both macrophages and neutrophils are recruited early upon *Salmonella* infection. Tg(*mfap4:mCherry-F/mpx:GFP*) larvae were injected with either PBS or Sal-E2Crimson in hindbrain ventricle. Time-lapse videos of labeled cells were acquired by confocal microscopy at 3 hpi during 12 hr and image series were collected every 1 hr. Two representative movies (maximum projections) of PBS-injected larva (left panel) and *Salmonella*-infected larva (right panel) are presented, showing recruitment of both neutrophils (green) and macrophages (red) to the infection site (*Salmonella*, magenta). Time is indicated in the top left corner of each panel. Scale bar: 30 µm.

https://elifesciences.org/articles/89828/figures#video4

To image long-term *Salmonella* infection, we used a *Salmonella* strain harboring a stable chromosomal version of the *GFP* gene (Sal-GFP-int). Macrophage reporter embryos Tg(*mfap4:mCherry-F*) were infected with Sal-GFP-int and imaged at 4 dpi using intravital confocal microscopy. Infected larvae displayed a massive accumulation of macrophages in the HBV, with clusters associated with persisting fluorescent *Salmonella* (*Figure 4A*, shown with arrows). 3D reconstruction of *mfap4*-positive cell volumes from confocal images revealed that *Salmonella* resided mainly inside some macrophages and only few *Salmonella* outside macrophages were observed (*Figure 4B*). Similar experiments with neutrophil reporter embryos, Tg(*mpx:GFP*), showed that although persisting bacteria could be observed, few neutrophils were present in the HBV at 4 dpi, but they were not clustered and did not contain persistent *Salmonella* (*Figure 4C, D*).

To simultaneously visualize the relative position of both macrophages and neutrophils at the infection site with persistent *Salmonella*, Tg(*mfap4:mCherry-F/mpx:GFP*) embryos were infected with Sal-E2Crimson and imaged at 4 dpi

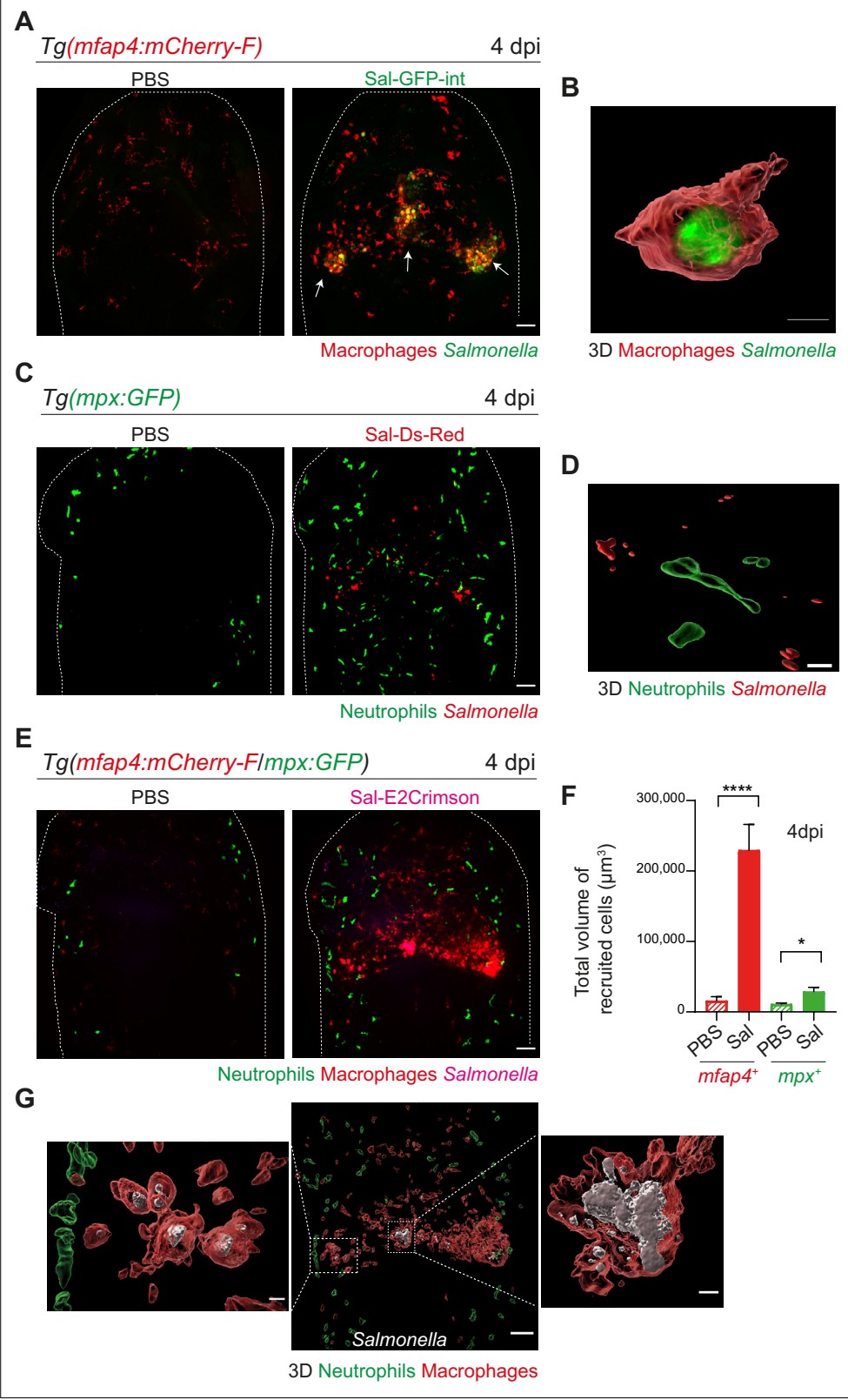

**Figure 4.** *Salmonella* hindbrain ventricle (HBV) infection induces hyper-accumulation of macrophages harboring persistent bacteria at late time point of infection. (**A**) Tg(*mfap4:mCherry-F*) larvae were injected with either PBS or Sal-GFP-int in HBV. Representative maximum projections of fluorescent confocal images, showing accumulation of macrophages (red) that co-localize with persistent *Salmonella* (green) at 4 dpi. Scale bar: 50 μm. (**B**) 3D

*Figure 4 continued on next page*

*Figure 4 continued*

reconstruction of persistent *Salmonella* residing inside a macrophage at 4 dpi. Scale bar: 5 µm. (**C**) Tg(*mpx:GFP*) larvae injected with either PBS or Sal-DsRed in HBV. Representative maximum projections of fluorescent confocal images showing that *Salmonella* (red) do not co-localize with neutrophils (green) at 4 dpi. Scale bar: 50 µm. (**D**) 3D reconstruction of neutrophils and persistent *Salmonella* at 4 dpi. Scale bar: 30 µm. (**E–G**) Tg(*mfap4:mCherry-F/mpx:GFP*) larvae were injected with either PBS or Sal-E2Crimson in HBV. (**E**) Representative maximum projections of fluorescent confocal images showing macrophage clusters (red), persistent *Salmonella* (magenta) and neutrophils (green) at 4 dpi. Scale bar: 50 µm. (**F**) Quantification of the total volume of recruited cells (*mfap4*[+] or *mpx*[+] cells) at 4 dpi. Data of three replicates pooled (mean volume/larva ± SEM, $n_{Sal}$ = 20, $n_{PBS}$ = 8, Mann–Whitney test, two-tailed, *p < 0.05, ****p < 0.0001). (**G**) 3D reconstruction of the HBV (middle panel) showing macrophage clusters (red) in which *Salmonella* (gray) persist, surrounded by neutrophils (green) at 4 dpi. Right and left panels are zooms of regions boxed by dotted lines. Scale bar: 30 µm. Scale bar zooms: 10 µm.

(*Figure 4E* and *Video 5*). Quantification of the total volume of recruited macrophages (*mfap4*[+] cells) and neutrophils (*mpx*[+] cells) in HBV confirmed an important macrophage recruitment, which was five time stronger at 4 dpi than at 16 hpi (*Figures 3H and 4F*). In contrast, neutrophils were similarly recruited at 4 dpi compared to 16 hpi. Importantly, this experiment confirmed that neutrophils do not co-localize with persistent *Salmonella* at late stages of infection (*Figure 4G*). In contrast, bacteria were primarily associated with macrophages at 4 dpi (*Figure 4G*).

Moreover, we quantified the proportion of macrophages harboring bacteria, revealing that the frequencies of infected macrophages within the recruited macrophage population ranged from 20% to 30% in the early stages to around 60% during the late stages (4 dpi) (*Figure 5A*). Intracellular bacterial levels of infected leukocytes were also compared between these two stages (*Figure 5B*). At 5 hpi, the volumes of E2Crimson-positive events (E2Crimson[+]) were lower than that at 4 dpi (*Figure 5C*). The size distribution analysis of E2Crimson[+] events indicated a higher representation of smaller sizes (0.5–1.5 and 1.5–10 µm³) at 5 hpi compared to 4 dpi, a stage during which very large E2Crimson[+] events were observed (100–1000 µm³, with some exceeding 1000 µm³) (*Figure 5D*). This analysis supports an elevated number of bacteria within macrophages during persistent stages and that intracellular bacteria are predominantly observed as clusters rather than as single cells.

Altogether, these findings show that at later stages of infection, *Salmonella* infection triggers a massive macrophage recruitment in the HBV with formation of large macrophage clusters containing numerous persistent bacteria.

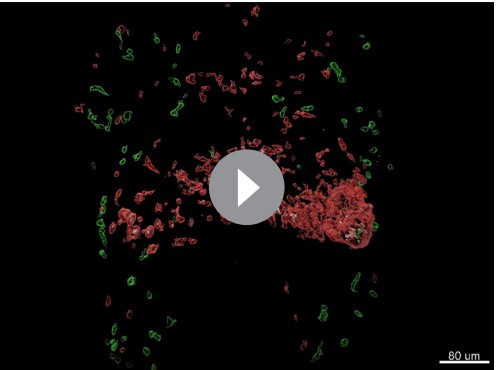

**Video 5.** *Salmonella* late infection induces hyper-accumulation of macrophages harboring persistent bacteria. Tg(*mfap4:mCherry-F/mpx:GFP*) larvae were injected with either PBS or Sal-E2Crimson in hindbrain ventricle and imaged by confocal microscopy at 4 dpi. This movie is a 3D reconstruction animation created from representative fluorescent confocal images using Imaris software and shows neutrophils (*mpx*[+] cells, green) and a cluster of macrophages (*mfap4*[+] cells, red) containing persistent *Salmonella* (white) at 4 dpi. Scale bar: 100 µm.

https://elifesciences.org/articles/89828/figures#video5

## During *Salmonella* infection, macrophages first acquire a M1-like phenotype and polarize toward non-inflammatory phenotype at later stages

The initial macrophage response to *Salmonella* infection was previously shown to be pro-inflammatory in mouse and zebrafish models (*Eisele et al., 2013*; *Pham et al., 2020*; *Hall et al., 2013*; *Ordas et al., 2011*; *Sheppe et al., 2018*). We thus hypothesized that zebrafish macrophages polarize toward M1-like phenotype upon *Salmonella* infection. The pro-inflammatory cytokine TNFa is a known marker for M1-like macrophage in various species including zebrafish (*Nguyen-Chi et al., 2015*). First, the transgenic line Tg(*tnfa:GFP-F*) was used to track Tnfa producing cells by expression of a farnesylated GFP (GFP-F) during acute infection. Embryos were infected with Sal-DsRed and imaged every 5 min from 1 to 10 hpi. Time-lapse videos showed that GFP-F expression was induced at the infection site and labeled cells

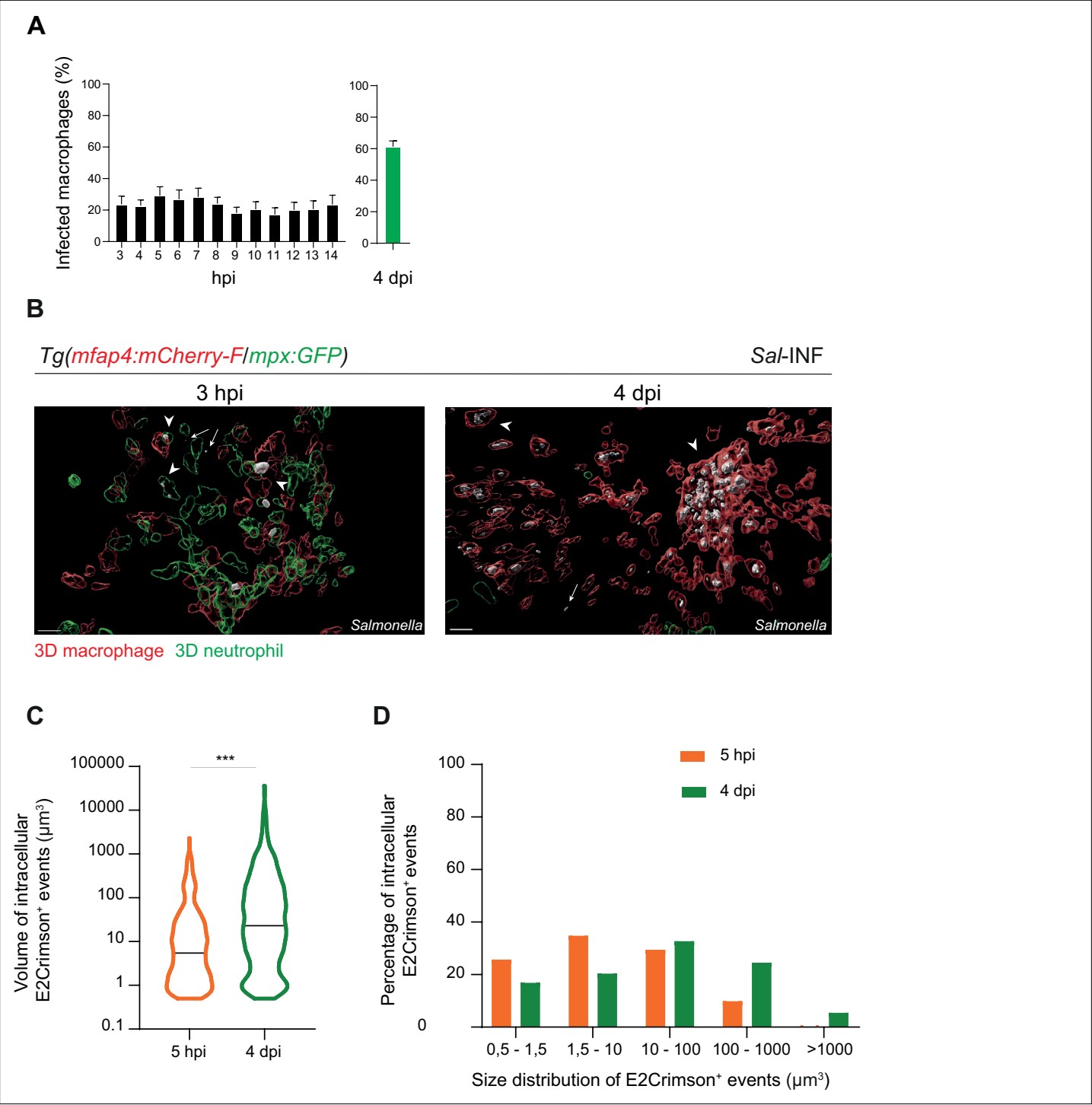

**Figure 5.** *Salmonella* persistence leads to a higher proportion of macrophages containing bacteria. Tg(*mfap4:mCherry-F/tnfa:GFP-F*) and Tg(*mfap4:mCherry-F/mpx:GFP*) larvae were injected with PBS or Sal-E2Crimson in HBV. (**A**) Quantification of the percentage of infected macrophages in *Sal*-infected larvae at indicated time post-infection. Data of three replicates pooled (mean percentage/larva ± SEM, $n_{3-14\,hpi}$ = 15 embryos and $n_{4\,dpi}$ = 43 embryos). (**B**) 3D reconstruction confirming the intramacrophagic and intraneutrophilic localization of a bacterial aggregate at 3 hpi and 4 dpi. Scale bar: 20 μm. (**C**) Size of intracellular E2Crimson+ events (in μm³), quantified following 3D reconstruction. Median volume, $n_{5\,hpi}$ = 506, $n_{4\,dpi}$ = 990, Mann–Whitney test, two-tailed, ***p < 0.001. (**D**) Size repartition of E2Crimson+ events (10 embryos were imaged at each time point).

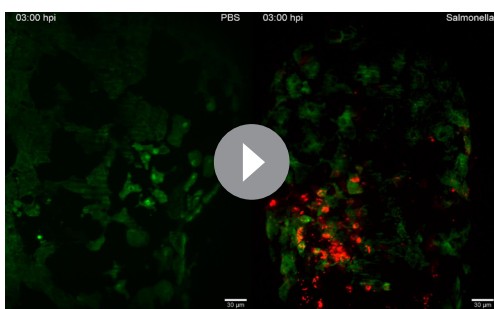

**Video 6.** *Salmonella* early infection induces a strong activation of tumor necrosis factor a (tnfa) -expressing cells. Tg(*tnfa:GFP-F*) larvae were injected with PBS or Sal-DsRed in the HBV. Time-lapse videos of labeled cells were acquired using light sheet microscopy at 3 hpi during 8 hr and image series were collected every 5 min. Two representative movies (maximum projections) of PBS-injected larva (left panel) and *Salmonella*-infected larva (right panel) are presented, showing recruitment of *tnfa*⁺ cells (green) to the infection site (*Salmonella*, red). Time is indicated in the top left corner of each panel. Scale bar: 30 µm.
https://elifesciences.org/articles/89828/figures#video6

harbored a myeloid morphology, suggesting that they are macrophages (*Figure 6—figure supplement 1A* and *Video 6*). To investigate the dynamic of macrophage polarization during *Salmonella* infection in vivo, we used the double transgenic line Tg(*mfap4:mCherry-F/tnfa:GFP-F*) in which all macrophages express a farnesylated mCherry and cells producing Tnfa, express GFP-F (*Nguyen-Chi et al., 2015*). Tg(*mfap4:mCherry-F/tnfa:GFP-F*) larvae were infected with Sal-E2Crimson and imaged at early time and at 4 dpi (*Figure 6* and *Video 7*). Quantification of volumes of *mfap4-* and *tnfa*-positive cells, every hour from 4 to 15 hpi, showed that *Salmonella* infection induced a strong *tnfa* response increasing over time, unlike PBS-injected controls in which only few *tnfa*-positive cells were observed (*Figure 6A–C* and *Figure 6—figure supplement 1A*). During the first hours of infection, a maximum of 80% of *mfap4*-positive cells became also *tnfa*-positive (*mfap4*⁺*tnfa*⁺ cells) corresponding to M1-like activated macrophages (*Figure 6B*). Interestingly, both macrophages with and without bacteria expressed *tnfa* (*Figure 6C*). These results show that in the early phase of *Salmonella* infection, most of the macrophages polarize toward M1 like, including both infected and bystander macrophages.

To check macrophage polarization in persistently infected zebrafish, we analyzed infected Tg(*mfap4:mCherry-F/tnfa:GFP-F*) larvae at 4 dpi (*Figure 6D*). Macrophage volume analysis revealed that few *tnfa*-positive macrophages (*mfap4*⁺*tnfa*⁺ cells) were still present at the infection site (less than 10%), but that the vast majority of accumulated macrophages were *tnfa*-negative (*Figure 6D–F* and *Figure 6—figure supplement 1B, C*). In agreement with the finding that the vast majority of accumulated macrophages at 4 dpi were not expressing *tnfa*, persistent *Salmonella* were found within *tnfa*-negative macrophages at late stages of infection (*Figure 6E*).

These results reveal that recruited macrophages switch their phenotype from M1-like states toward non-M1 states during the time course of *Salmonella* infection and that *Salmonella* survives in non-M1 macrophages at late stages of infection.

## Macrophages display dynamic transcriptional profiles upon *Salmonella* infection

To interrogate the molecular basis of macrophage activation during *Salmonella* infection, we compared the transcriptomic profiles of 'activated' macrophages in infected host versus 'inactivated' macrophages in non-infected condition at 4 hpi and 4 dpi. First, we confirmed that Fluorescence-Activated Cell Sorting (FACS) from Tg(*mfap4:mCherry-F*) allowed enrichment in macrophage population, as shown by RT-qPCR analysis of *mfap4* mRNA (*Figure 7—figure supplement 1*). Second we designed the following experimental setup: Tg(*mfap4:mCherry-F/tnfa:GFP-F*) larvae were either infected with *Salmonella* (*Sal*-INF) or not infected (Non-INF) (*Figure 7A*). FACS analysis on whole larvae in Non-INF condition revealed that a large majority of *mfap4*⁺ cells (mCherry⁺) were *tnfa*⁻ (GFP⁻) both at 4 hpi and 4 dpi (100% and 90.4% ± 4, respectively) (*Figure 7—figure supplement 2*). These expected populations were referred to as 'inactivated' macrophages. FACS analysis in Sal-INF condition at 4 hpi revealed that the majority (92.5% ± 6) of *mfap4*⁺ cells (mCherry⁺) were *tnfa*⁺ (GFP⁺), while the majority (80.9% ± 4) were *tnfa*⁻ (GFP⁻) at 4 dpi (*Figure 7—figure supplement 2*). Therefore, *mfap4*⁺ *tnfa*⁺ cells at 4 hpi, which represent the main macrophage population of early infection phase, were referred to as 'M1-activated'. In contrast the main macrophage population at 4 dpi consists of *mfap4*⁺ *tnfa*⁻ cells that were referred to as 'non-M1-activated'.

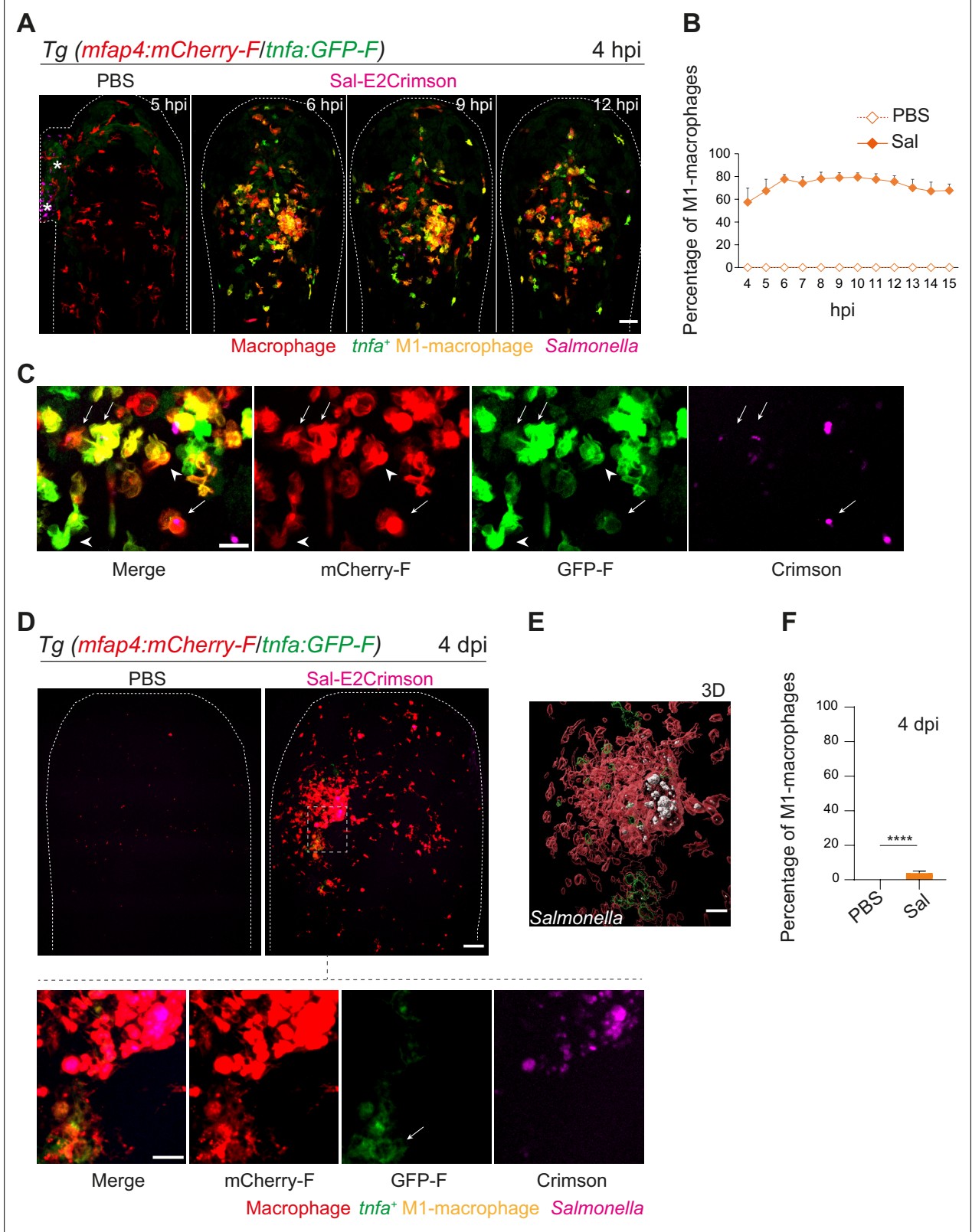

**Figure 6.** Macrophages polarize toward a pro-inflammatory M1-like phenotype upon *Salmonella* infection at early stage but not at late stage. (**A–F**) Tg(*mfap4:mCherry-F/tnfa:GFP-F*) larvae were injected with PBS or Sal-E2Crimson in HBV. (**A**) Representative maximum projections of fluorescent confocal images extracted from a 4D sequence, showing recruitment of macrophages (*mfap4+* cells, red) and M1-like activation (*mfap4+-tnfa+* cells, yellow) to *Salmonella* (magenta) from 4 to 15 hpi. Asterisk: auto-fluorescence. Scale bar: 50 μm. (**B**) Quantification of the percentage of M1 macrophages

*Figure 6 continued on next page*

*Figure 6 continued*

at indicated time post-infection. Data of two replicates pooled (mean percentage/larva ± SEM, *n* = 12 per condition). (**C**) Zoom of fluorescent confocal images in A. Scale bar: 20 µm, arrow: infected *tnfa*⁺ macrophages and arrowhead *tnfa*⁺ bystander macrophages. (**D**) Representative maximum projections of fluorescent confocal images of PBS-injected and Sal-E2Crimson-infected larvae at 4 dpi (upper panels). Scale bar: 50 µm. Zooms of regions boxed by dotted lines (bottom panels). Scale bar zoom: 10 µm. (**E**) 3D reconstruction of macrophage clusters (red) containing persistent *Salmonella* (gray), surrounded by few *tnfa*⁺ macrophages (green) at 4 dpi. Scale bar: 10 µm. (**F**) Quantification of the percentage of M1 macrophages at 4 dpi. Data of four replicates pooled (mean percentage/larva ± SEM, 4 dpi, $n_{Sal}$ = 23 larvae, $n_{PBS}$ = 20, one sample Wilcoxon test, ****p < 0.0001).

The online version of this article includes the following figure supplement(s) for figure 6:

**Figure supplement 1.** *Salmonella* early infection induces a strong activation of tumor necrosis factor (TNF)-expressing cells.

Global gene expression analysis on these different populations was performed by RNA sequencing (***Figure 7A***). The principal component analysis showed that each biological sample formed distinct clusters according to the 4 hpi and 4 dpi experimental conditions (***Figure 7B***). All were pure myeloid populations as all sorted populations expressed considerable levels of several key macrophage markers such as *mfap4*, *mpeg1.2*, *marco*, *csf1ra*, and *lygl1,* but neither muscle marker (*smyhc2*), lymphocyte markers (*lck*, *rag2*, and *cd79b*), neutrophil marker (*mpx*) nor other cell type markers (***Figure 7C*** and ***Figure 7—figure supplement 3***). In addition, only the cells sorted as *mfap4*⁺ *tnfa*⁺ from the Sal-INF condition at 4 hpi expressed *tnfa* (***Figure 7C***). To compare the functional differences between 'inactivated' and 'activated' macrophages, we performed a differentially expressed gene (DEG) analysis between Non-INF and Sal-INF conditions at 4 hpi and 4 dpi (adjusted p value (p-adj) <0.05|log₂(Fold Change (FC)) ≥1). *Salmonella* infection induced massive changes of gene expression in macrophages at 4 hpi (***Figure 7D***, ***Figure 7—source data 1***) and M1-activated macrophages harbored specific gene expression signature with 3173 up-regulated and 2847 down-regulated genes, whereas at 4 dpi, only 104 up-regulated and 440 down-regulated genes were specific to the non-M1 signature (***Figure 7E, F*** and ***Figure 7—source data 1***). These results reveal that macrophages display a highly dynamic signature during the time course of infection.

At 4 hpi, M1-activated macrophages were characterized by a specific up-regulation of pro-inflammatory genes including those involved in the TNF pathway (*tnfa*, *traf4b*, *traf7*, *traf1b*, *tnfrsf1b*, *tnfrsf9b*, *tnfaip2*, and *tnfaip3*), pro-inflammatory cytokines (*il12a* and *il12bb*), chemotaxis (*ccl38a.5*, *ccl38a.4*, *ccl19*, *cxcl18b*, and *cxcl20*), complement (*c3a.1*, *c3a.3*, *c3a.4*, and *c3a.6*), innate immune sensing of pathogens and antimicrobial responses (*tlr5*, *stat1a*, *stat2*, *mxc*, *irf9*, *crfb17*, *card11*, *card9*, and *hif1ab*), and matrix metalloproteinases (*mmp9*, *mmp14a*, *mmp14b*, *mmp11b*, *mmp13b*, *mmp30*, and *mmp23bb*) (***Figure 6F***). Interestingly, *il10* and *il6*, that were described for their dual role during inflammation (***Mocellin et al., 2003***; ***Scheller et al., 2011***), were also up-regulated, while known markers of M2 and anti-inflammation such as *mrc1b*, *tgm2l*, *tgfbr2*, *klf2*, *klf4*, *irf2*, and *mertka* were down-regulated. Up-regulated genes at 4 hpi were classified according the gene ontology (GO) terms and the KEGG pathways (***Figure 7—figure supplement 4B***, ***Figure 7—source data 1***). Top ranking GO terms enriched in macrophages at 4 hpi were 'inflammatory response', 'neutrophil migration', 'macrophage migration', and 'defense response to gram negative bacterium'. Enriched KEGG pathways were 'Protein processing in endoplasmic reticulum', 'phagosome', and 'cytokine–cytokine receptor interactions' (***Figure 7G***,

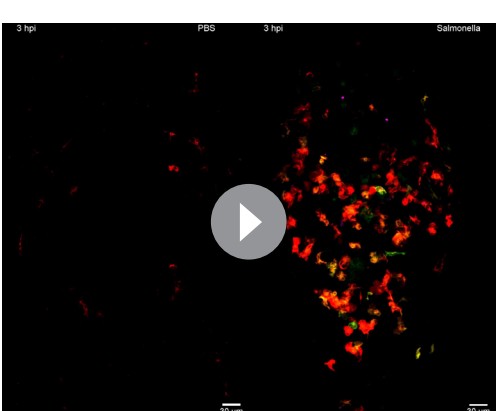

**Video 7.** Macrophages polarize toward a pro-inflammatory M1-like phenotype upon *Salmonella* early infection. Tg(*mfap4:mCherry-F/tnfa:GFP-F*) larvae were injected with PBS or Sal-E2Crimson in the HBV. Time-lapse videos of labeled macrophages were acquired using confocal microscopy at 3 hpi during 12 hr and image series were collected every 1 hr. Two representative movies (maximum projections) of PBS-injected larva (left panel) and *Salmonella*-infected larva (right panel) are presented, showing recruitment of macrophages (*mfap4*⁺ cells, red) and M1-like activation (*mfap4*⁺-*tnfa*⁺ cells, yellow) to *Salmonella* (magenta). Time is indicated in the top left corner of each panel. Scale bar: 30 µm.

https://elifesciences.org/articles/89828/figures#video7

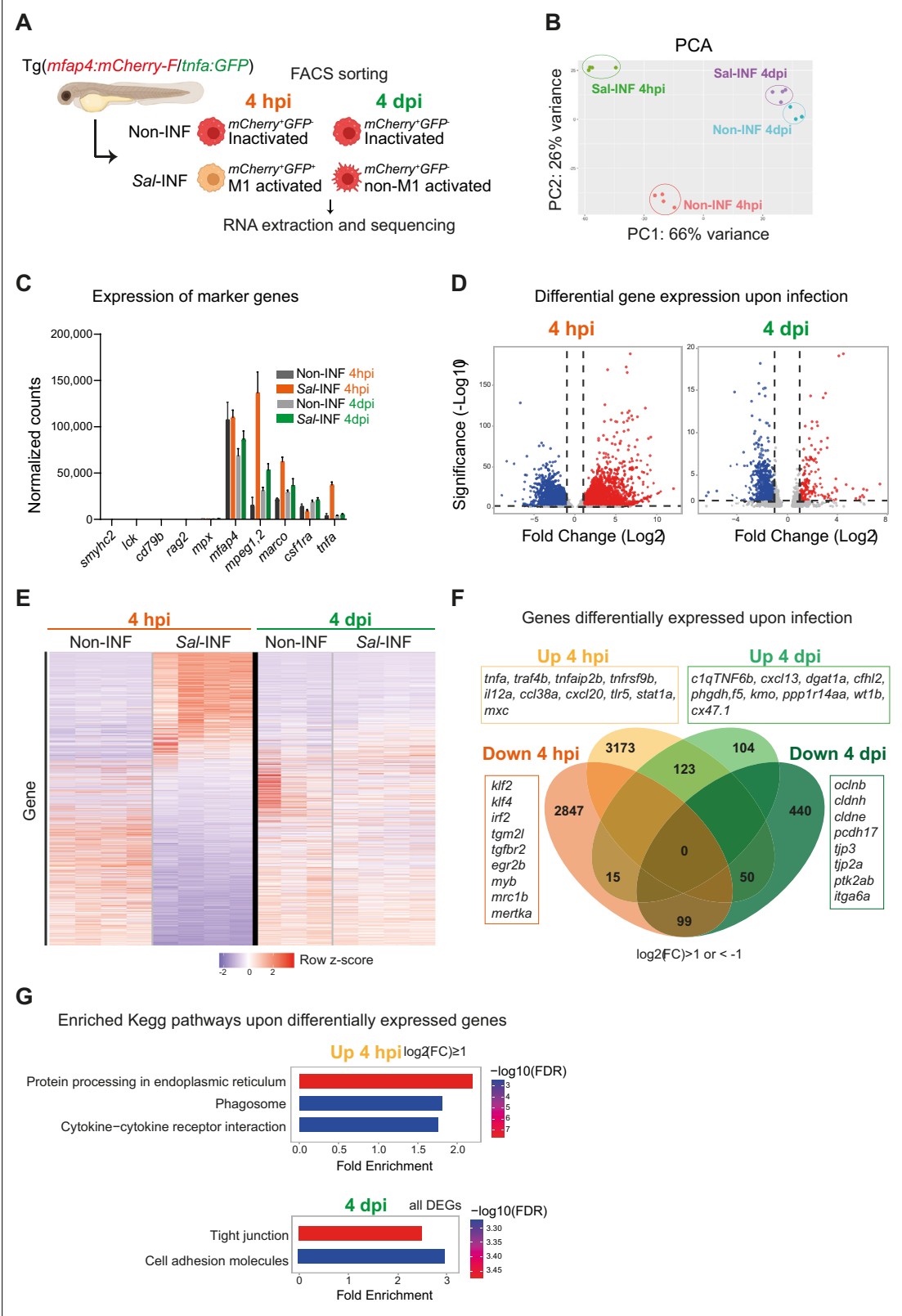

**Figure 7.** RNAseq analysis reveals macrophage transcriptome switch during *Salmonella* infection. (**A**) Schematic diagram of macrophage RNA-sequencing experimental design. Tg(*mfap4:mCherry-F/tnfa:GFP-F*) larvae were either infected with *Salmonella* (*Sal*-INF) or non-infected (Non-INF). Fluorescence-Activated Cell Sorting (FACS) was used to isolate *mfap4*⁺-*tnfa*⁻ cells (mCherry⁺ GFP⁻) and *mfap4*⁺-*tnfa*⁺ cells (mCherry⁺ GFP⁺) at 4 hpi and 4 dpi. (**B**) Principal component analysis (PCA) score plot of *mfap4*⁺-*tnfa*⁻ cells in Non-INF condition (*n* = 4) and *mfap4*⁺-*tnfa*⁺ cells in *Sal*-INF condition (*n*

*Figure 7 continued on next page*

*Figure 7 continued*

= 4) at 4 hpi and of *mfap4*⁺-*tnfa*⁻ cells in Non-INF condition (*n* = 3) and *mfap4*⁺-*tnfa*⁻ cells in *Sal*-INF condition (*n* = 4) at 4 dpi. (**C**) Normalized expression of several marker genes of muscle cells, lymphocytes, neutrophils, and macrophages in the different sorted macrophage populations. (**D**) Volcano plot showing differentially expressed genes (DEGs) between Non-INF and *Sal*-INF conditions at 4 hpi and 4 dpi. Adjusted p value (p-adj) <0.05 was used as the threshold to judge the significance of the difference in gene expression. Red plots: up-regulated genes; blue plots: down-regulated genes; gray plots: unchanged genes. (**E**) Heatmap of DEGs between macrophage populations across infection (p-adj <0.05|Log₂(FC) ≥1). Selected top DEGs from each population are shown. Color coding, decreased expression: blue, no expression: white, high expression: red. (**F**) Venn diagram showing unique and intersecting up- or down-regulated genes (indicated as Up and Down, respectively) upon infection from macrophage transcriptome at 4 hpi and 4 dpi. The numbers of up- and down-regulated genes are indicated in bold in each unique and overlapping sector of the Venn diagram. The most noteworthy genes of each unique sector of the Venn diagram are indicated (p-adj <0.05|(Log₂(FC) ≥1 or ≤1)). (**G**) Chart representation of Kyoto Encyclopedia of Genes and Genomes (KEGG) pathways enriched in up-regulated genes (p-adj <0.05|Log₂(FC) ≥1) at 4 hpi (upper panel) and all DEGs at 4 dpi (lower panel) (p-adj <0.05). Graph shows the fold enrichment, red color: lowest enrichment false discovery rate (FDR) and blue color: highest enrichment FDR.

The online version of this article includes the following source data and figure supplement(s) for figure 7:

**Source data 1.** List of all the differentially expressed genes upon *Salmonella* infection and analysis.

**Source data 2.** List of selected differentially expressed genes upon *Salmonella* infection.

**Figure supplement 1.** *mfap4* mRNA expression in mCherry-F⁺ or mCherry-F⁻ sorted cells.

**Figure supplement 2.** Macrophage populations were sorted by Fluorescence-Activated Cell Sorting (FACS) before transcriptomic analysis.

**Figure supplement 3.** Normalized expression of marker genes confirms the purity of sorted macrophage populations.

**Figure supplement 4.** RNAseq analysis shows dynamic transcriptional profiles of macrophages upon *Salmonella* infection.

*Figure 7—source data 1*). Together, these results confirm the pro-inflammatory signature of *tnfa*⁺ macrophages at 4 hpi.

In contrast non-M1-activated macrophages at 4 dpi specifically up-regulated genes involved in anti-inflammation and M2 markers in mammalian systems (*c1qtnf6b*, *cxcl13*, *dgat1a*, *cfhl2* [ortholog of *F13B* in human], *phgdh*, *f5*, *kmo*, and *ppp1r14aa*), retinoic acid pathway (*rdh10a*, *rdh12*, and *ugt1a*), bacterial defense (*npsn*, *c9*, *hamp*, and *nos2b*), regeneration (*wt1b*), tissue protection and development (*cx47.1*, *lta*, and *ecm1b*), and extracellular matrix remodeling (*mmp25b*) (*Figure 7F*, *Figure 7— source data 1*). Interestingly, some pro-inflammatory genes, like *saa*, *cxcl11.1*, *s100b*, *ptgs2a*, *itgb3b*, *illr4* (*CLEC* orthologous), and *pppr2r2bb* were up-regulated at both times points but with a higher fold change at 4 hpi than 4 dpi, while others like *noxo1a*, *lsp1*, *plat*, *lpar1*, *selm*, *elf3*, and *irf6* were up-regulated at 4 hpi and down-regulated at 4 dpi. In contrast, genes involved in anti-inflammatory response and tissue regeneration (*prg4a*, *cxcl19*, *cd59*, and *clic2*) were down-regulated at 4 hpi and up-regulated at 4 dpi (*Figure 7—figure supplement 4A*). Of note, some markers of microglia (*apoeb*, *ccl34b.1*, and *csf1rb*) were down-regulated at both time points, suggesting a re-differentiation of this cell population upon HBV infection. KEGG pathways analysis performed on DEGs at 4 dpi revealed the absence of a pro-inflammatory signature and that the most significantly enriched pathways were 'tight junction' and 'cell adhesion' (*Figure 7G*, *Figure 7—source data 1*), suggesting profound changes in their polarization status and cell-adhesion program. Surprisingly, the most significantly enriched GO terms in up-regulated genes included 'Astrocyte', 'Myelination', 'Oligodendrocyte', and 'Myelin maintenance' (*Figure 7—figure supplement 1C*, *Figure 7—source data 1*). Macrophages may express genes supporting astrocyte and oligodendrocyte function in the injured microenvironment of the infected HBV or these up-regulated genes may reflect the presence of internalized transcripts after efferocytosis in the infected area, as previously observed in other biological context (*Lantz et al., 2020*).

We confirmed the specific expression profile for one candidate gene via an alternative approach. *Wilms Tumor 1b* (*wt1b*) delineates a pro-regenerative macrophage subset and was shown to be required for heart and fin regeneration in zebrafish (*Sanz-Morejón et al., 2019*). We checked the expression of *wt1b* in macrophages by crossing Tg(*wt1b:GFP*) line with Tg(*mfap4:mCherryF*) line to track simultaneously *wt1b*-expressing cells and macrophages. Double transgenic embryos were infected and imaged at 4 hpi and 4 dpi. Confocal microscopy analysis showed that, while macrophages did not express *wt1b* at 4 hr after infection, they strongly expressed *wt1b* 4 days after infection. Large clusters of *wt1b*-positive macrophages harboring persistent bacteria were also observed (*Figure 8*). These data reveal phenotypical distinct macrophage signatures during early and persistent

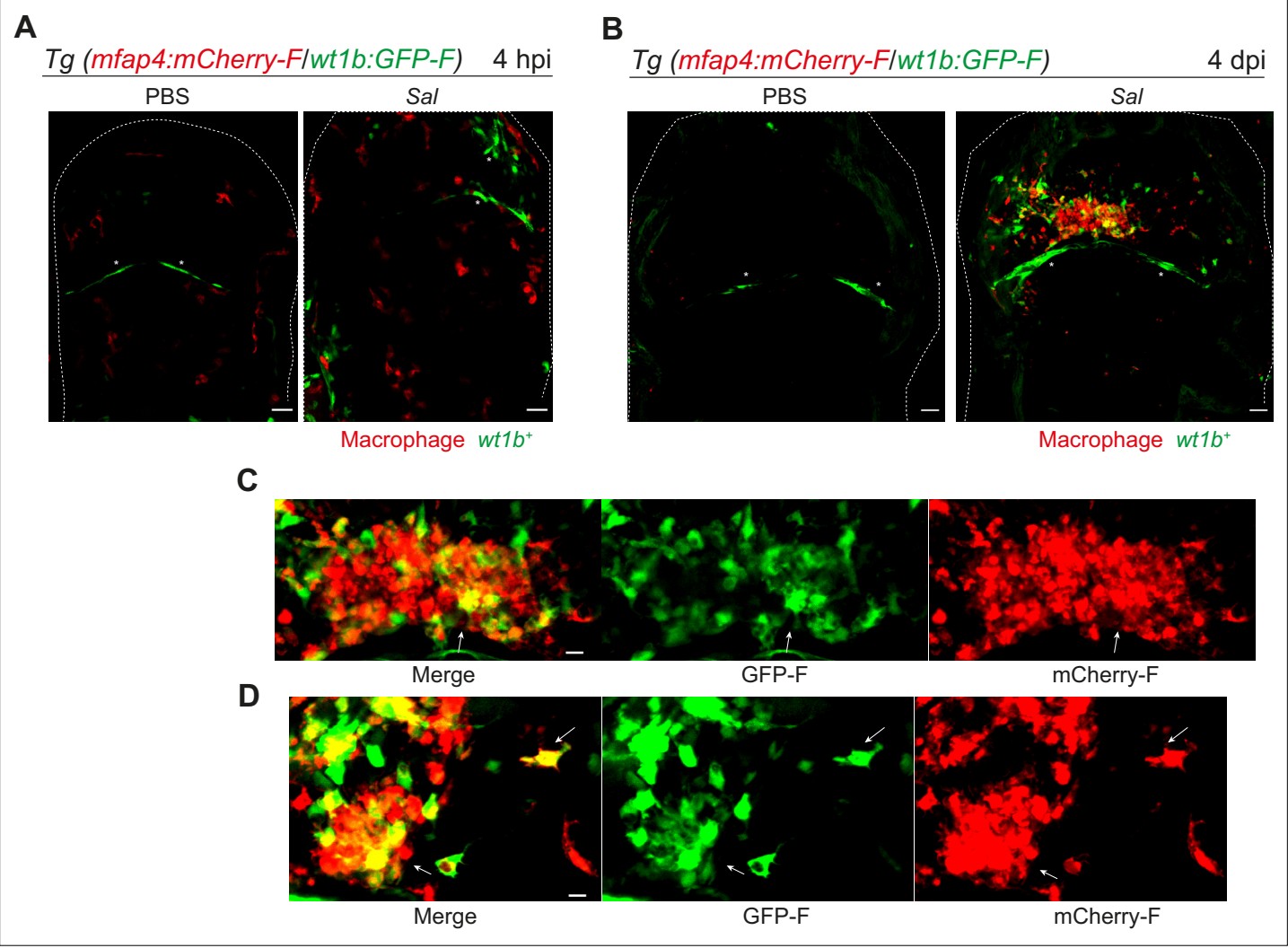

**Figure 8.** *Salmonella* persistence induces large clusters of pro-regenerative *wt1b*-expressing macrophages. (**A–D**) Tg(*mfap4:mCherry-F/wt1b:GFP-F*) larvae were injected with PBS or Sal-E2Crimson in the HBV. (**A**) Representative maximum projections of fluorescent confocal images showing macrophages (*mfap4*+ cells, red) and *wt1b*-expressing cells (*wt1b*+ cells, green) after *Salmonella* infection at (**A**) 4 hpi and at (**B**) 4 dpi. Asterisk: auto-fluorescence. Scale bar: 30 μm. Big clusters of *wt1b*-expressing macrophages (*mfap4*+-*wt1b*+ cells, yellow) were observed at 4 dpi. (**C, D**) Zoom of two representative fluorescent confocal images showing large clusters of *wt1b*-expressing macrophages (*mfap4*+-*wt1b*+ cells, yellow) at 4 dpi. Scale bar: 10 μm, arrow: mfap4+-wt1b+ macrophage.

infection stages that can be identified in situ and highlight the pro-regenerative state of late non-M1-activated macrophages.

Altogether, these data suggest that macrophages skew their phenotype from acute to persistent infection, switching from a pro-inflammatory phenotype to an anti-inflammatory/pro-regenerative phenotype with a unique cell-adhesion signature.

### *Salmonella* persistence in the host is accompanied with macrophages harboring low motility

Our transcriptomic analysis data revealed that many genes related to cell adhesion were differentially regulated in *Salmonella* infection compared to control and that this cell-adhesion signature was not the same during acute and persistent infections (*Figure 9A*). These genes encompassed cell–cell-adhesion molecules, likes cadherins (*cdh1*, *cdh2*, *cdh11*, *cdh17*, and *cdh18*), occludins (*oclna*, *oclnb*, *cldnk*, and *cln19*) and tight junction proteins (*tjp1a*, *tjp3*, *tjp2a*, and *tjp2b*), extracellular matrix-adhesion proteins (*itgb1a*, *itgb1b*, *itgb4*, *itga6a*, and *cd44a*), regulators of cell adhesion (*pxna*, *ptk2ba*,

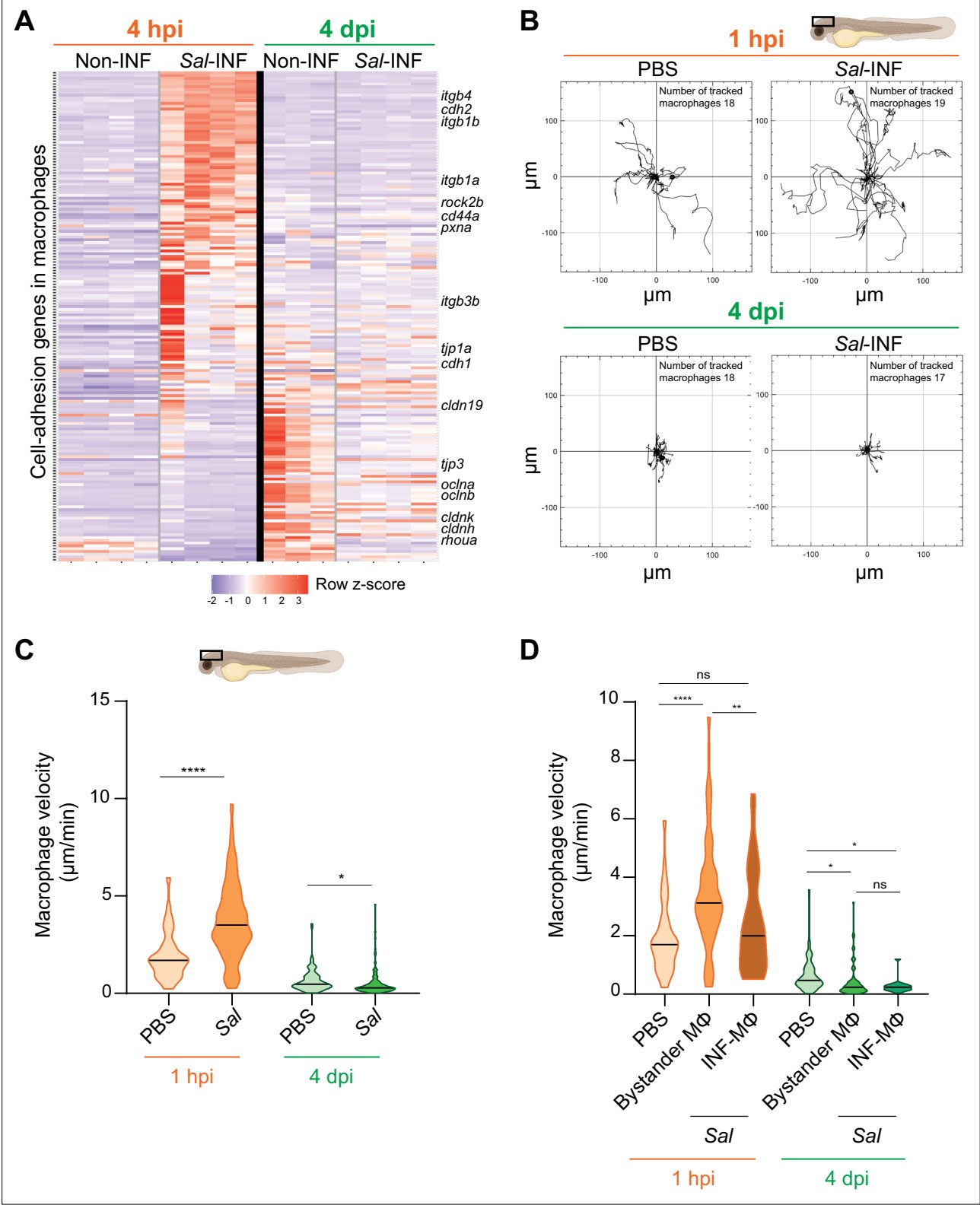

**Figure 9.** *Salmonella* persistence induces drastic changes in cell adhesion-related gene expression and motility in macrophages. (**A**) Heatmap of differentially expressed genes (DEGs) involved in cell adhesion, between macrophage populations across infection (p-adj <0.05). Selected DEGs from each population are indicated. Color coding, decreased expression: blue, no expression: white, high expression: red. (**B–D**) Tg(*mfap4:mCherry-F*) larvae were injected with PBS or Sal-E2Crimson in HBV and time-lapse videos of labeled macrophages were acquired during 2 hr at 1 hpi or 4 dpi. (**B**)

*Figure 9 continued on next page*

*Figure 9 continued*

Migration of macrophages in response to PBS or Sal-E2Crimson at 1 hpi or 4 dpi. Representative trajectory plots of individual macrophage movement tracks are shown, with the initial position in the center of the graph. Number of macrophage tracks are indicated. (**C**) Quantification of the individual macrophage velocity from PBS-injected or Sal-infected larvae at 1 hpi and 4 dpi. Data of four replicates per time point pooled (mean velocity/macrophage ± SEM, at 1 hpi: $n_{PBS}$ = 76, $n_{Sal}$ = 195; at 4 dpi: $n_{PBS}$ = 93, $n_{Sal}$ = 162; t-test, two-tailed, significance of Sal versus PBS conditions, ****p < 0.0001, *p < 0.05). (**D**) Quantification of the individual bystander macrophage or infected macrophage velocity from PBS-injected or Sal-infected larvae at 1 hpi and 4 dpi. Data of two replicates per time point pooled (mean velocity/macrophage ± SEM, at 1 hpi: $n_{PBS}$ = 76, $n_{bystander-M\Phi}$ = 92; $n_{infected-M\Phi}$ = 33; at 4 dpi: $n_{PBS}$ = 93, $n_{bystander-M\Phi}$ = 67; $n_{infected-M\Phi}$ = 17; analysis of variance (ANOVA) Kuskal–Wallis' test with Dunns' post-test, ****p < 0.0001, **p < 0.01, *p < 0.05, ns: not significant).

The online version of this article includes the following figure supplement(s) for figure 9:

**Figure supplement 1.** Neutrophils remain motile during the establishment of *Salmonella* persistent infection.

and *ptk2ab*), and regulators of cytoskeleton remodeling (*rac1a*, *rock2b*, and *rhoua*) and most of them are down-regulated at 4 dpi (*Figure 9A*). This shift in adhesion program of activated macrophages during the establishment of persistent infection suggests profound changes in their interaction with their environment and their migratory properties. Indeed, macrophage function relies on the establishment of contact with neighboring cells and stable attachments to their substrate, allowing transmigration through tissues, positioning, and signaling. Cadherins were shown to regulate macrophage functions (*To et al., 2022*; *Van den Bossche et al., 2015*) and integrins and small GTPases are also important for macrophage migration (*Aflaki et al., 2011*; *Paterson and Lämmermann, 2022*). To test whether the changes in expression of cell adhesion and cytoskeletal regulators were mirrored by changes in macrophage adhesion associated motility, we generated a double transgenic line, Tg(*mfap4:Gal4/UAS:nfsB-mCherry*), in which *mfap4* promoter drives indirectly the expression of NTR fused with mCherry specifically in macrophages. In these larvae, macrophages are strongly and mosaically labeled by red fluorescence allowing an accurate tracking of individual macrophages. Larvae were infected with Sal-E2Crimson and imaged using confocal microscopy during 2 hr at 1 hpi and at 4 dpi. Analysis of macrophage trajectories at the injection site revealed that macrophage motility at 1 hpi was enhanced in infected larvae, with long-distance migration compared to PBS control (*Figure 9B* and *Videos 8, 9*). In contrast, macrophage motility at the injection site at 4 dpi in infected larvae was decreased compared to PBS control, with very

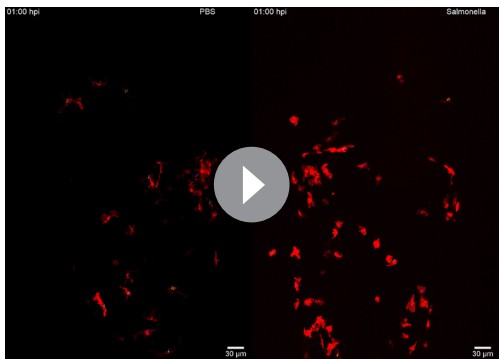

**Video 8.** Highly motile macrophages are recruited during *Salmonella* early infection. Tg(*mfap4:mCherry-F*) larvae were injected with PBS or Sal-E2Crimson in HBV. Time-lapse videos of labeled macrophages were acquired by confocal microscopy during 2 hr at 1 hpi and image series were collected every 2 min. Two representative movies (maximum projections) of PBS-injected larva (left panel) and *Salmonella*-infected larva (right panel) are presented, showing in different colors individual macrophage trajectory tracks (*mfap4*+ cells, red). Time is indicated in the top left corner of each panel. Scale bar: 30 µm.

https://elifesciences.org/articles/89828/figures#video8

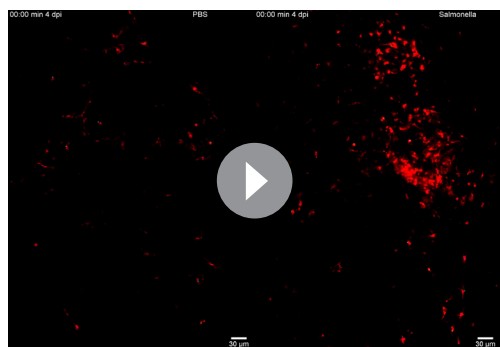

**Video 9.** The motility of macrophages is decreased *Salmonella*-injected larvae compared to PBS-injected larvae during persistent infection. Tg(*mfap4:mCherry-F*) larvae were injected with PBS or Sal-E2Crimson in HBV. Time-lapse videos of labeled macrophages were acquired confocal microscopy during 2 hr at 4 dpi and image series were collected every 2 min. Two representative movies (maximum projections) of PBS-injected larva (left panel) and *Salmonella*-infected larva (right panel) are shown, showing in different colors individual macrophage trajectory tracks (*mfap4*+ cells, red). Time is indicated in the top left corner of each panel. Scale bar: 30 µm.

https://elifesciences.org/articles/89828/figures#video9

short trajectories. In addition, macrophage migration speed was significantly enhanced at 1 hpi in the infected larvae compared to the control condition, whereas at 4 dpi, this speed was reduced compared to the control condition (*Figure 9C*). To compare the motility of uninfected bystander macrophages versus infected macrophages, a similar analysis was performed on macrophages with and without intracellular bacteria. At 1 hpi, bystander macrophages exhibited heightened velocity during acute infection, while infected macrophages maintained a speed comparable to the control group (*Figure 9D*). In contrast, during persistent infection (4 dpi), both bystander and infected macrophages exhibited reduced velocity compared to unstimulated macrophages (*Figure 9D*). To evaluate the motility of another leukocyte population, we measured neutrophil velocity. At 1 hpi, neutrophil speed was increased by the infection while at 4 dpi, it was unchanged compared to control (*Figure 9—figure supplement 1*). Importantly, neutrophil velocity was higher than that of macrophages in each condition. These observation underscores the dynamic cellular characteristics exhibited by leukocytes, including macrophages, influenced by their infection status, and the stage of the infection. These data also demonstrate that, during persistent infection, clusters of motionless macrophages are formed while they provide a niche for *Salmonella* to survive.

## Discussion

While some pathogens transiently infect an organism, others can survive inside the host for a long period of time or even for life. *Salmonella* can persist inside macrophages for long term (*Monack, 2012*; *Gal-Mor, 2019*) and this persistence has been proposed to depend on macrophage polarization status (*Eisele et al., 2013*). Here, we describe the first model of *Salmonella* persistence in transparent zebrafish that allows the detailed analysis of the dynamic interactions between *Salmonella* and polarized macrophages during early and late phases of infection in a whole organism.

In other zebrafish infection models with *Salmonella*, acute symptoms – hyper-proliferation of the bacteria, larval mortality – presumably prevent the establishment of a long-term infection (*Stockhammer et al., 2009*; *van der Sar et al., 2003*). Unlike these studies, we injected *Salmonella* in a closed compartment, the larval HBV. This mode of infection led to diverse outcomes, including acute infection, clearance, and persistence. Indeed, in 47% of the larvae, *Salmonella* replicated intensively, leading to the rapid larval death. In surviving larvae, the bacteria were either cleared or survived inside the host, possibly up to 14 dpi. These bacteria that survive for an extended period of time within the tissues were called 'persistent *Salmonella*'. These 'persistent bacteria' should not be confused with 'Persisters', which refers to bacteria that persist for prolonged periods despite antibiotic treatment (*Fisher et al., 2017*), even though they may share some common features. These diverse outcomes of infected zebrafish could be explained by multiple factors, including stochastic cell-to-cell differences in genetically identical bacteria or the complexity of immune cell population. Indeed, in in vitro systems, heterogeneous activity of the bacteria creates phenotypically diverse bacterial subpopulations which shape different cellular environments and potentiate adaptation to a new niche (*Saliba et al., 2016*; *Avraham et al., 2015*).

We focused our in vivo study on embryos infected with persistent *Salmonella* up to 4 dpi. During the course of infection, the infected host displayed two main phases characterized by distinct inflammatory status: an early pro-inflammatory phase and a late phase characterized by both pro- and anti-inflammatory signals. We analyzed the interaction of *Salmonella* with innate immune cells during both phases. Similar to previous work (*Masud et al., 2019*), during early infection, both neutrophils and macrophages engulfed the bacteria. Macrophages played a crucial role in bacterial clearance, as shown by the increased bacterial burden upon macrophage depletion. Exploiting the imaging possibilities of this system, we visualized macrophage activation thanks to *tnfa* expression in real time in response to *Salmonella* infection and showed that engulfing macrophages and bystanders polarized toward M1-like phenotypes. This pro-inflammatory response from the host was rapid and robust as more than 90% of the macrophages activated as M1 like within the first hours. Besides, we assessed the full transcriptome of zebrafish macrophages during early infection and we demonstrated that macrophages adopted a pro-inflammatory program characterized by the up-regulation of genes involved in inflammation, cytokine–cytokine receptor, and phagosome pathways. These results are consistent with our initial analysis on whole larvae and are reminiscent to previous studies using whole *Salmonella*-infected larvae where genes encoding matrix metalloproteinases, pro-inflammatory cytokines, and chemokine pathways are up-regulated (*Stockhammer et al., 2009*; *Ordas et al., 2011*).

There are also similarities with the activation program detected in human macrophages infected with different pathogens including *Salmonella* (*Nau et al., 2002*), which may represent an evolutionarily conserved program for host defense.

During the persistent phase, high-resolution intravital imaging showed that *Salmonella* mainly localized inside macrophages that were organized in large clusters. In contrast, although some neutrophils were detected in the HBV, they were poorly recruited, not clustered and did not harbor persistent bacteria. Importantly, most of the mobilized macrophages did not express *tnfa* in late stages of infection; we could not observe infected macrophages expressing *tnfa*. Transcriptomic profiling of these non-inflammatory macrophages during bacterial persistence revealed an anti-inflammatory and regenerative profile characterized (1) by the attenuation of pro-inflammatory genes and (2) by the up-regulation of genes involved in anti-inflammation, tissue maintenance, development, and regeneration. This included the expression of *wt1b* an important pro-regenerative gene that defines a pro-regenerative macrophage subtype in zebrafish (*Sanz-Morejón et al., 2019*). Surprisingly, neuronal function-related genes, involved in myelination or expressed in astrocytes and oligodendrocytes were also up-regulated. This should not result from cell contamination given the macrophage specificity of the RNAseq datasets, which did not recover other cell type markers. We therefore propose two possible scenarios. In the first scenario, the brain damages caused by the persistent infection may trigger a regenerative response involving pro-regenerative macrophages, which participate to the regeneration of axons and the restoration of their function by remyelination and maintenance of myelin homeostasis (*McNamara et al., 2023*; *Rawji et al., 2016*; *Tsarouchas et al., 2018*). Alternatively, during persistent infection, macrophages may perform efferocytosis of damaged or dying cells and internalize transcripts originating from them. The internalization of apoptotic RNAs by macrophages during efferocytosis has been demonstrated in the context of murine macrophages co-cultured with apoptotic human T cells (*Lantz et al., 2020*). Both scenarios suggest that the macrophages interact tightly with their microenvironment during persistent infection. It is important to emphasize that the dynamic variations in macrophage transcriptomes between acute and persistent infections may indicate disparities in proliferation, death, and polarization, altering the composition of macrophage populations over time. However, it might also denote a phenotypic switch from M1- to M2-like polarization within individual macrophages over time. In the future, a more comprehensive analysis at single-cell level of resolution will better discriminate phenotypically distinct populations of macrophages at play in this model.

Our data indicate that Non-M1 macrophages which have limited bactericidal activity, can be used by *Salmonella* as a niche to sustain persistent infections in their hosts. *Salmonella* persistence was previously associated with M2 macrophages in a mouse model of long-term infection (*Eisele et al., 2013*; *Goldberg et al., 2018*). Molecularly, M2-permissive macrophage polarization in granulomas is partially dependent on SteE, an effector of the *Salmonella* pathogenicity island-2 type III secretion system (SPI2 T3SS), while the host cytokine TNF limits M2 polarization (*Pham et al., 2020*). In infected cultured macrophages, SteE regulates STAT3, reorienting macrophage polarization toward M2 phenotypes (*Gibbs et al., 2020*; *Panagi et al., 2020*). Another study revealed that macrophages harboring *Salmonella* Typhimurium during persistent stages express high levels of the inducible nitric oxide synthase (known as iNOS or NOS2) (*Goldberg et al., 2018*). Interestingly, we also observed an up-regulation of *nos2b* in zebrafish macrophages 4 days after *Salmonella* infection. Furthermore, non-growing, antibiotic-tolerant bacteria, also called persisters, were shown to translocate SPI2 T3SS effectors, including SteE, into the macrophage to reprogram it into a non-inflammatory macrophage (*Stapels et al., 2018*). Our data are in line with these studies, emphasizing the potential of the zebrafish model for the study of persistent infections. However, further investigations are still needed to identify the bacterial factors involved in macrophage reprogramming in our system.

An efficient immune response requires the interplay of a cocktail of cell-adhesion molecules and immune cells (*Cui et al., 2018*; *Friedl and Weigelin, 2008*). Macrophages express various cell-adhesion proteins including *integrin b1* (*itgb1*) that is crucial for cell movements and protrusiveness during surveillance (*Paterson and Lämmermann, 2022*). Tight junctions are important for particle uptake and exchange (*Blank et al., 2011*) and RAC proteins were shown to be necessary for macrophage basic motility and migration (*Rosowski et al., 2016*; *Wheeler et al., 2006*). We demonstrated a shift in macrophage adhesion program during persistent stages with several cell adhesion-related genes that were down-regulated, including *itgb1*, *rac2*, and tight junction-related genes. This shift

in adhesion program was accompanied with a decrease in macrophage motility. The observation that macrophages remained stationary while neutrophil exhibited normal mobility implies that cell density within tissue does not appear to affect the motility of leukocytes in persistent infection stages. Chemokines also play a pivotal role in macrophage chemotaxis, and alterations in these signaling pathways may contribute to the lower motility of macrophages during persistent infection stages. Previous studies have established the involvement of Ccl2 and its receptor Ccr2 in guiding macrophages toward bacterial pathogens in mice and zebrafish (*Cambier et al., 2014*). Here, both *ccr2* and *ccl2* orthologs, *ccl38a.4*, were up-regulated at 4 hpi, yet their expression reverted to baseline levels by 4 dpi. In contrast, *Salmonella* infection had no impact on the expression of *cxcr4b* and *sdf1* (*cxcl12a*), representing another crucial chemoattractant signal. Based on our transcriptomic analysis, *cxcr3* and *cxcl11* are regulated in response to *Salmonella* infection. Therefore, our data suggest a dynamic shift in major chemotaxis signals between the acute and persistent stages of *Salmonella* infection that may contribute to the regulation of macrophage motility. Because these motionless macrophages appear non-inflammatory and contain persistent bacteria, they may constitute a permissive environment for bacterial survival. Of interest, the *Salmonella* SPI2 T3SS effector SseI, which is involved in long-term infection in mice, has been previously found to modulate the migration of cultured macrophages (*McLaughlin et al., 2009*). Subversion of macrophage motility has been also previously observed with another pathogen, *M. marinum*, which uses the ESX-1/RD1 secretion system to enhance macrophage recruitment to nascent granulomas and favor granulomas formation (*Davis and Ramakrishnan, 2009*).

Granulomas are a key pathological feature of some intracellular bacterial infections; they do contain a high proportion of macrophages. *Salmonella enterica* can cause granuloma formation in different animal species (*Work et al., 2019*), including mice (*Goldberg et al., 2018*) and humans (*Muniraj et al., 2015*; *Narechania et al., 2015*; *Nasrallah and Nassar, 1978*). The complex cellular structure of granuloma is thought to be important for bacterial persistence. Recently, using single-cell transcriptomics, Pham et al. identified diverse macrophage populations within *Salmonella* Typhimurium-induced granulomas (*Pham et al., 2023*). In the present study, *Salmonella* persisted within aggregates of *tnfa*-negative macrophages that remind early granulomas. Mycobacterial granulomas are characterized

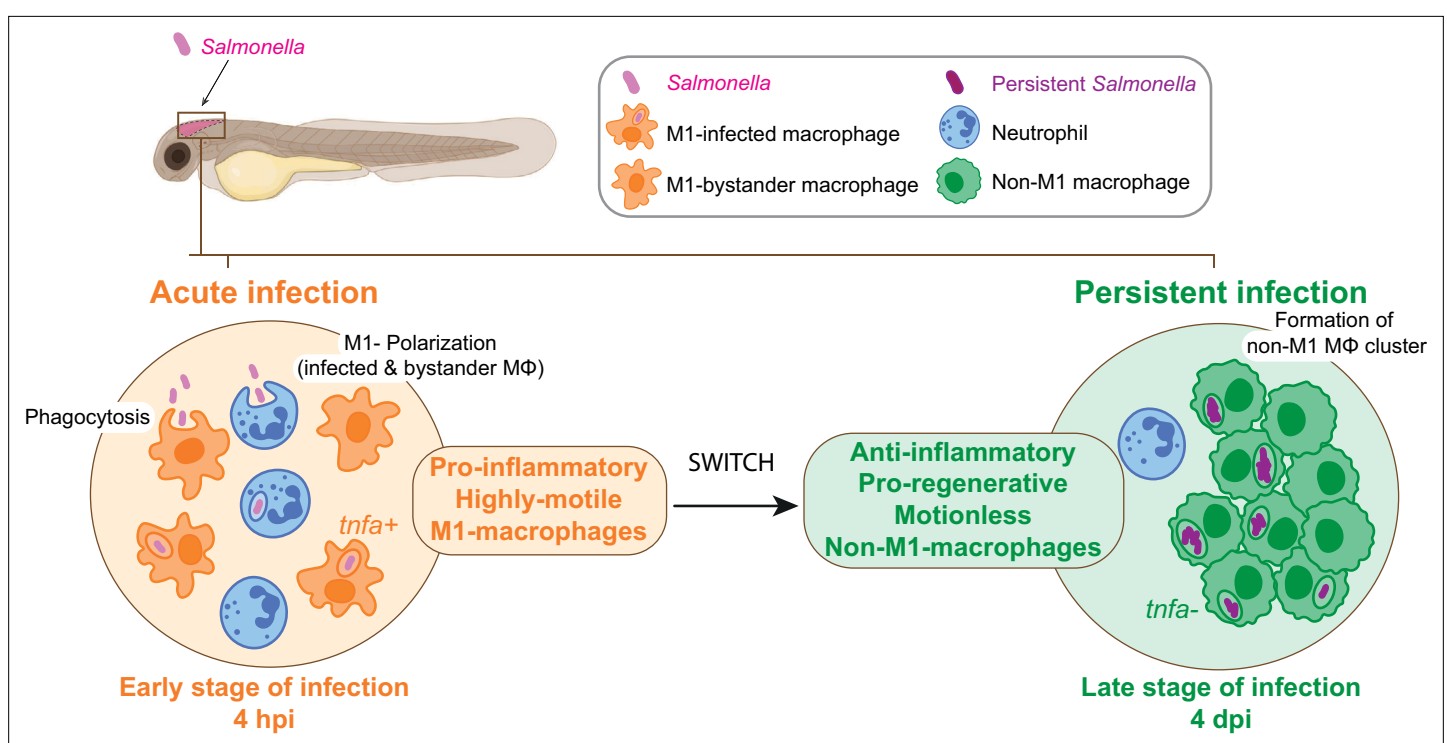

**Figure 10.** From acute to persistent *Salmonella* infection, macrophages switch their polarization states and motility. Schematic representation of the two main phases of *Salmonella* hindbrain ventricle infection in zebrafish. The early phase corresponds to an acute infection characterized by the recruitment of leukocyte populations, phagocytosis of *Salmonella* and M1 polarization of highly motile macrophages (MΦ), while during the late-phase *Salmonella* persists inside motionless macrophages (MΦ) that display an anti-inflammatory and pro-regenerative status and form clusters.

by epithelioid macrophages expressing tight junction proteins, E-cadherin (CDH1) and ZO-1 (*tjp1*) (*Cronan et al., 2021*; *Pagán and Ramakrishnan, 2018*). Interestingly, in the context of persistent *Salmonella* infection, we showed that *tnfa*-negative macrophages down-regulated the expression of several tight junction-related genes during persistence, such as *cdh1* and *tjp1a*. This is similar to what has been observed in *Salmonella*-induced granulomas in mice where macrophages weakly express E-cadherin or ZO-1 (*Goldberg et al., 2018*) and suggests that these structures are non-epithelioid granulomas initiated by an innate immune mechanism.

In conclusion, the zebrafish larva proves to be an extraordinary platform for the analysis of *Salmonella* persistent infection and the understanding of long-term interactions with host cells inside a whole living animal. The highly dynamic changes of macrophage gene expression between early infection and persistent phases confirm the highly versatile nature of macrophage-*Salmonella* interactions and suggest that macrophage polarization and motility switch plays an important role in the establishment of a secure niche for *Salmonella* (*Figure 10*).

## Materials and methods

### Fish husbandry
Fish (*Danio rerio*) maintenance, staging and husbandry were performed as described (*Nguyen-Chi et al., 2014*) with golden, AB strains and transgenic lines. *Tg(mfap4:mCherry-F)ump6tg* (*Phan et al., 2018*) referred to as *Tg(mfap4:mCherry-F)* and Tg(*mfap4:Gal4VP16)ump10TG* referred to as Tg(*mfap4:Gal4*) were used to visualize macrophages. *Tg(mpx:eGFP)ill4* referred to as *Tg(mpx:eGFP)* (*Renshaw et al., 2006*) was used to visualize neutrophils. *Tg(tnfa:GFP-F)ump5Tg* referred to as *Tg(tnfa:GFP-F)* was used to visualize cell expressing *tnfa* (*Nguyen-Chi et al., 2015*). *Tg(mfap4:mCherry-F)* crossed with *tg(tnfa:GFP-F)* were used to visualize activated M1 macrophages. *Tg(mpeg1:Gal4)gl25* (*Ellett et al., 2011*) and *Tg(UAS-E1b:nfsB-mCherry)i149* (*Davison et al., 2007*) were used to ablate macrophages. Tg(*wt1b:GFP*)li1Tg were used to visualize wt1b-expressing cells (*Bollig et al., 2009*). Embryos were obtained from pairs of adult fishes by natural spawning and raised at 28°C in tank water. Embryos and larvae were staged according to *Kimmel et al., 1995* and used for experiments from 0 hpf to 17 dpf. Larvae were anaesthetized in zebrafish water supplemented with 200 µg/ml tricaine (ethyl 3-aminobenzoate methanesulfonate, MS-222 Sigma #A5040) before any manipulation (infection or imaging) and if necessary were replaced in their medium at 28°C.

### *Salmonella* strains
*Salmonella* strains were grown overnight in Luria-Bertani (LB) medium at 37°C with 100 µg/ml ampicillin, 10 µg/ml tetracycline, or 25 µg/ml kanamycin when required. *Salmonella enterica* serovar Typhimurium ATCC14028s (here called *Salmonella*) was used as the original parental *Salmonella* strain. *Salmonella* carrying plasmid pRZT3::dsRED (*van der Sar et al., 2003*), pE2-Crimson (Clontech), and pFPV25.1 (*Valdivia and Falkow, 1996*), that express red fluorescent protein (dsRED), far-red fluorescent protein (E2Crimson), and green fluorescent protein (GFP), respectively, were used for microinjection in zebrafish embryos (see below). To create a *Salmonella* strain expressing chromosomal copies of GFP, the *rpsM::gfp* fusion from strain SM022 (*Vazquez-Torres et al., 1999*) was transferred by P22 transduction of the *rpsM::gfp* fusion, linked to kanamycin resistance gene into the original parental *Salmonella* strain ATCC14028s through selection for kanamycin resistance.

### *Salmonella* injections
*Salmonella* strains were grown to exponential phase and recovered by centrifugation, washed twice and resuspended in PBS at an $OD_{600}$ of 5 or 2.5 (depending on the required dose) with phenol red. Infection was carried out by microinjection of 1.5 nl of bacterial suspensions in the Hindbrain Ventricule (HBV) of dechorionated and anesthetized 2 dpf embryos. Two different doses of *Salmonella* were used for microinjection in zebrafish embryos: low (<500 CFU) and high (1000–2000 CFU). The inoculum dose was checked by counting the CFU containing in 1.5 nl of the bacterial suspension. For larva survival analysis, a minimum of 30 larvae were infected per replicate and three replicates were done for each experiment.

## Quantification of bacterial load by CFU counts

A minimum of five larvae per time points were anesthetized in zebrafish water supplemented with 200 µg/ml tricaine and then each embryo was crushed in 1% Triton X-100-PBS in an Eppendorf tube using a pestle (MG Scientific #T409-12). After 10 min incubation at room temperature, dilutions of total lysates were plated on LB agar plates containing appropriate antibiotics. CFU were counted after an overnight incubation of the plates at 37°C. Larvae used for CFU counts were randomly chosen among surviving larvae.

## Macrophage ablation

For macrophage depletion, we used Tg(*mpeg1:Gal4/UAS:nfsB-mCherry*) embryos expressing *nfsB-mCherry* under the indirect control of *mpeg1* promoter. *nfsB-mCherry* encodes an *Escherichia coli* nitroreductase (NTR) fusionned to mCherry protein that converts Metronidazole (MTZ) into a toxic agent that kills the cells. Tg(*mpeg1:Gal4/UAS:nfsB-mCherry*) embryos were incubated in zebrafish water containing 10 mM MTZ (Sigma-Aldrich) and 0.1% DMSO at 48 hpf and 24 hr before injection with *Salmonella* or PBS. Treatment with 0.1% DMSO was used as a control. Depletion efficiently was assessed by imaging using the MVX10 Olympus microscope just before HBV injection. Effects of macrophage depletion on embryo survival and bacterial load during infection were analyzed at 1, 2, 3, and 4 dpi.

## Generation of the macrophage reporter line, Tg(*Mfap4:Gal4VP16*)

The Gal4VP16 ORF was amplified by PCR and used to replace the mCherry ORF downstream of the Mfap4 promoter in the transgenesis vector used in *Phan et al., 2018* to generate the Tg(*zMfap4:mCherry-F*)ump6TG insertion. The resulting plasmid, Tol2zMfap4:Gal4VP16 was co-injected with the ISce-I meganuclease in fertilized Tg(*mpx:gal4/UAS:nfsB-mcherry*) eggs. The offspring with red fluorescent macrophages and no fluorescent neutrophils were raised and screened for transmission of the Tg(*Mfap4:Gal4VP16*) insertion to its offspring.

## Imaging of live zebrafish larvae

Larvae were anesthetized in 200 µg/ml tricaine, positioned on 35 mm glass-bottomed dishes (WillCo-dish), immobilized in 1% low-melting-point agarose and covered with 2 ml of embryo water supplemented with 160 µg/ml tricaine. Epi-fluorescence microscopy was performed by using an MVX10 Olympus MacroView microscope that was equipped with MVPLAPO ×1 objective and XC50 camera. Confocal microscopy was performed using an ANDOR CSU-W1 confocal spinning disk on an inverted NIKON microscope (Ti Eclipse) with ANDOR Neo sCMOS camera (×20 air/NA 0.75 objective). Image stacks for time-lapse acquisitions were performed at 28°C. The 4D files generated by the time-lapse acquisitions were processed using ImageJ as described below. Three-dimensional reconstructions were performed on the four-dimension files for time-lapse acquisitions or on three-dimension files using Imaris (Bitplan AG, Zurich, Switzerland). In *Figure 3C* and in *Figure 6—figure supplement 1*, a custom-made LSFM developed at ICFO was used (*Bernardello et al., 2022*). For the described LSFM experiments, we made use of two illumination air objectives (Nikon ×4/NA 0.13), one water-immersion detection objective (Olympus ×20/NA 0.5), a sCMOS camera (Hamamatsu Orca Flash4.v2), and a 200-mm tube lens (Thorlabs), obtaining an overall magnification of ×22.2. For LSFM imaging, zebrafish embryos were embedded within a fluorinated ethylene propylene (FEP) tube (ID 2 mm, OD 3 mm) containing 0.2% low melting agarose (LMP) agarose with the addition of 160 µg/ml tricaine. The inner walls of the FEP tube would have been previously coated with a 3% methyl cellulose layer to avoid tail adhesion. After plugging with 1.5% LMP agarose the bottom end of the FEP tube, it was inserted into the LSFM imaging chamber and mounted vertically and upside-down. The temperature-controlled chamber was then filled with 15 ml of embryo water supplemented with 160 µg/ml tricaine.

## Visualization of interaction between *Salmonella*, macrophages, and neutrophils

The 3D and 4D files generated by confocal microscopy were processed using ImageJ. First, stacks of images from multiple time points were concatenated and then brightness and contrast were adjusted for better visualization with the same brightness and contrast per channel for every infected and PBS-control larva in each experiment. To generate the figure panels, stacks of images were compressed

into maximum intensity projections and cropped. For visualization of bacteria localization related to macrophages and neutrophils, surfaces tool of Imaris software was used to reconstruct the 3D surfaces of leukocytes and bacteria.

## Quantification of total leukocyte population, quantification of recruited neutrophils and macrophages, and intracellular bacterial aggregates

To quantify total leukocyte populations, transgenic reporters were tricaine-anesthetized and whole larvae were imaged using MVX10 Olympus microscope. Total numbers of fluorescent leukocytes were counted by computation using Fiji (ImageJ software) as following: (1) leukocytes (Leukocyte Units, LU) were detected using 'Find Maxima' function, (2) maxima were automatically counted using run ('ROI Manager…'), roi-Manager ('Add'), and (3) roiManager ('Measure') functions. To quantify recruited leukocyte populations in the HBV, only the infection sites of reporter larvae were imaged using Spinning disk Nikon Ti Andor CSU-W1 microscope. ImageJ was used to concatenate stacks of images containing multiple time points and to adjust and set the same brightness and contrast per channel for every infected and PBS-control larva in each experiment. 'Surfaces' tool of Imaris was used to reconstruct in 3D cell surfaces. Total volumes of immune cells were then extracted for relative quantification and expressed as mean volume ($\mu m^3$) with standard error of the mean. For the quantification of the percentage of M1 macrophages, the ratio of the total volume of *mfap4*+-*tnfa*+ cells among *mfap4*+ cells was calculated. To quantify intracellular bacterial aggregates in the HBV, 'Surfaces' tool of Imaris was used to reconstruct in 3D surfaces based on E2Crimson fluorescence. Only intracellular E2Crimson-positive events were analyzed. Volumes of each E2Crimson-positive event were then extracted.

## Leukocyte tracking and motility analysis

Macrophages and neutrophils were tracked in every time step using Manual Tracking ImageJ PlugIn. Velocity was extracted directly from manual tracking data table. Directionality graphs were obtained by Chemotaxis and Migration Tool PlugIn using the X–Y position of each macrophage extracted from manual tracking data table.

## RNA preparation on whole larva and quantitative RT-PCR analysis

For quantitative RT-PCR analysis, 2 dpf larvae were either injected in the HBV with Sal-GFP or with PBS as described above. To determine the relative expression of *il1b*, *tnfb*, *tnfa*, *il8*, *ccr2*, *ccl38a.4*, *mmp9*, *mrc1b*, *nrros*, *mfap4*, *mpeg1*, *cxcr4b*, and *cxcl12a*, total RNA from infected larvae or controls (pools of eight larvae each) was prepared at 3 hpi, 1 dpi, 2 dpi, 3 dpi, and 4 dpi using Macherey-Nagel Nucleospin RNA Kit (# 740955.250). RNAs were reverse transcribed using OligodT and M-MLV reverse transcriptase (# 28025-013) according to the manufacturer's recommendations. Quantitative RT-PCR were performed using LC480 and SYBER Green (Meridian BIOSCIENCE, SensiFAST SYBR # BIO-98050) according to the manufacturer's recommendations and analyzed using LC480 software. The final results are displayed as the fold change of target gene expression in infected condition relative to PBS-control condition, normalized to *ef1a* as reference gene (mean values from eight independent experiments with standard error of mean) with the formula $2^{-\Delta\Delta CT}$. Results for *tnfa*, *cxcr4b*, and *cxcl12a* are presented as relative target gene expression in PBS or in infected condition, normalized to *ef1a* as reference gene ($2^{-\Delta CT}$). The primers used are listed below.

| Gene | Primers | Sequences 5'–3' |
|---|---|---|
| *ef1a* | Forward | TTCTGTTACCTGGCAAAGGG |
| | Reverse | TTCAGTTTGTCCAACACCCA |
| *il1b* | Forward | TGGACTTCGCAGCACAAAATG |
| | Reverse | CGAAGAAGGTCAGAAACCCA |
| *tnfa* | Forward | TTCACGCTCCATAAGACCCA |
| | Reverse | CCGTAGGATTCAGAAAAGCG |
| *tnfb* | Forward | CGAAGAAGGTCAGAAACCCA |
| | Reverse | GTTGGAATGCCTGATCCACA |

*Continued on next page*

*Continued*

| Gene | Primers | Sequences 5'–3' |
|------|---------|-----------------|
| *il8* | Forward<br>Reverse | CCTGGCATTTCTGACCATCAT<br>GATCTCCTGTCCAGTTGTCAT |
| *mmp9* | Forward<br>Reverse | CTCAGAGAGACAGTTCTGGG<br>CCTTTACATCAAGTCTCCAG |
| *ccr2* | Forward<br>Reverse | TGGCAACGCAAAGGCTTTCAGTGA<br>TCAGCTAGGGCTAGGTTGAAGAG |
| *ccl38a.4* | Forward<br>Reverse | GCATCTTCATCGCCTGTC<br>GCATCCACCAGATTCATCAG |
| *mrc1b* | Forward<br>Reverse | CGCCAAAGTGATGAGCCCAACT<br>GCAGGAAGCGATGTTGTGACCTT |
| *nrros* | Forward<br>Reverse | CTGTCCGTCGTGCTCAGTCA<br>GAGCTGACGACCGCTGCAC |
| *mfap4.2* | Forward<br>Reverse | GGAGGATGGACGGTGATTC<br>TCCTCCAGATCCACTCTCAGC |
| *mpeg1.1* | Forward<br>Reverse | GTGAAAGAGGGTTCTGTTACA<br>GCCGTAATCAAGTACGAGTT |
| *cxcr4b* | Forward<br>Reverse | GCACCACAAGTCCATTGCCA<br>GCTGTGAGAGGAGGGCGGTT |
| *cxcl12a* | Forward<br>Reverse | GCACACCTCCTTGTTGTTCTTC<br>TCCACAGTCAACACAGTCC |

## Transcriptomic analysis on FACS-sorted macrophages

Double transgenic larvae, *Tg(mfap4:mCherry-F; tnfa:GFP-F)*, were either non-infected (uninfected groups) or infected with non-labeled *Salmonella* at 48 hpf (infected groups), as described above. Cell dissociation from pools of 300 larvae were performed at 4 hpi and 4 dpi. Because *Salmonella* induces different outcomes in zebrafish larvae at late time points (4 dpi), it may result in different transcriptomic profiles of macrophages. We thus anticipated that bulk RNA sequencing would not provide meaningful biological signals and would be difficult to interpret. Therefore, we selectively focused on the 'infected' cohort, which displays persistent infection at 4 dpi, and only larvae harboring macrophage clusters in the brain were kept. Cell dissociation, FACS sorting, and RNA preparation were performed as described in *Begon-Pescia et al., 2022*. A total of 15 samples were processed for transcriptome analysis using cDNA sequencing. The four experimental groups are: 'uninfected/4 hpi', '*Salmonella* infected/4 hpi', 'uninfected/4 dpi', and '*Salmonella* infected/4 dpi'. Experimental groups were obtained from four replicates, except for the condition 'uninfected/4 dpi' which was obtained from three replicates. The 15 samples were sent to Montpellier GenomiX plateform (MGX, Institut de Genomique Fonctionnelle, Montpellier, France) for library preparation and sequencing. RNA-seq libraries were generated from 5 ng of RNA with the SMART-Seq v4 Ultra Low Input RNA Kit from Takara Bio (#634889) and the DNA Prep Kit from Illumina (#20060060) and were clustered and sequenced using an Illumina NovaSeq 6000 instrument, a flow cell SP and NovaSeq Xp Workflow according to the manufacturer's instructions, with a read length of 100 nucleotides. Image analysis and base calling were done using the Illumina NovaSeq Control Software and Illumina RTA software. Demultiplexing and trimming were performed using Illumina's conversion software (bcl2fastq 2.20). The quality of the raw data was assessed using FastQC (v11.9) from the Babraham Institute and the Illumina software SAV (Sequencing Analysis Viewer). FastqScreen (v0.15) was used to identify potential contamination. The RNAseq data were mapped on the zebrafish genome (version GRCz11) and gene counting was performed with Featurecounts v2.0.3. Sequencing depth of all samples was between 53 and 87 million reads. All reads were aligned to the GRCz11 version of the zebrafish genome using TopHat2 v2.1.1 (using Bowtie v2.3.5.1) software, and Samtools (v1.9) was used to sort and index the alignment files. Subsequently, normalization and differential gene expression analysis was performed using DESeq2 and edgeR methods. After comparison of the two methods, DESeq2 method (*Love et al., 2014*) was kept. P values were adjusted using *Benjamini and Hochberg, 1995* corrections for controlling false-positive rate (False Discovery Rate_FDR, also called p-adjusted (p-adj)) and results

were considered statistically significant when p-adj <0.05. Analysis was performed based on the $\log_2$ fold change (FC). Gene Ontology analysis of DEGs and KEGG pathways enrichment analysis were performed with ShinyGO 0.76 (*Ge et al., 2020*) with criteria of p -adj<0.05 up-regulated‖$\log_2$(FC) >0 or ≥1 and down-regulated‖$\log_2$(FC) <0 or ≤1. Purity of the different sorted macrophage populations was confirmed by absence of expression of several marker genes of various cell types (*Farnsworth et al., 2020*; *Metikala et al., 2021*).

## Statistical analysis

Studies were designed to generate experimental groups of approximatively equal size, using randomization and blinded analysis. The sample size estimation and the power of the statistical test were computed using GPower. A preliminary analysis was used to determine the necessary sample size $N$ of a test given $\alpha < 0.05$, power = $1 - \beta > 0.80$ (where $\alpha$ is the probability of incorrectly rejecting H0 when it is in fact true and $\beta$ is the probability of incorrectly retaining H0 when it is in fact false). Then the effect size was determined. Groups include the number of independent values, and the statistical analysis was done using these independent values. No inclusion/exclusion criteria of data were applied. The number of independent experiments (biological replicates) is indicated in the figure legends when applicable. The level of probability $p < 0.05$ constitutes the threshold for statistical significance for determining whether groups differ. GraphPad Prism 7 Software (San Diego, CA, USA) was used to construct graphs and analyze data in all figures. Specific statistical tests were used to evaluate the significance of differences between groups (the test and p value is indicated in the figure legend).

## Acknowledgements

This work was undertaken with support from Catherine Gonzalez and Victor Goulian from the Aquatic model facility ZEFIX from LPHI (University of Montpellier) and the qPCR Haut Debit platform of the University of Montpellier. We acknowledge the imaging facility BioCampus Montpellier Ressources Imagerie (MRI), member of the national infrastructure France-BioImaging supported by the French National Research Agency (ANR-10-INSB-04, 'Investments for the future') and Elodie Jublanc (Biocampus, Montpellier) for her help with microscopy. We thank Dr. Ferric C Fang (University of Washington School of Medicine) for *Salmonella* strain SM022, Dr Laurent Manchon for his advices for the RNAseq analysis and Pr Christoph Englert (Fritz Lipmann Institute) for transgenic line *Tg(wt-1b:GFP)*. ICFO authors also acknowledge financial support from the Spanish Ministry of Economy and Competitiveness through the 'Severo Ochoa' program for Centres of Excellence in R&D (CEX2019-000910-S), from Fundacio Privada Cellex, Fundacio Mir-Puig, Generalitat de Catalunya through the CERCA program, and Laserlab-Europe EU-H2020 (871124). MGX acknowledges financial support from France Génomique National infrastructure, funded as part of 'Investissement d'Avenir' program managed by Agence Nationale pour la Recherche (contract ANR-10-INBS-09).

## Additional information

### Funding

| Funder | Grant reference number | Author |
| --- | --- | --- |
| Horizon 2020 Framework Programme | MSCA-ITN ImageInLife Grant Agreement n° 721537 | Pablo Loza-Alvarez Georges Lutfalla |
| Horizon 2020 Framework Programme | MSCA-ITN Inflanet Grant Agreement n° 955576 | Mai E Nguyen-Chi |
| Agence Nationale de la Recherche | ANR-19-CE15-0005-01, MacrophageDynamics | Mai E Nguyen-Chi |
| Region Occitanie | REPERE « INFLANET » | Mai E Nguyen-Chi |

| Funder | Grant reference number | Author |
|---|---|---|
| Spanish Ministerio de Economía y Competitividad | CEX2019-000910-S | Matteo Bernardello<br>Emilio J Gualda<br>Pablo Loza-Alvarez |
| MINECO/FEDER | RYC-2015-17935 | Matteo Bernardello<br>Emilio J Gualda<br>Pablo Loza-Alvarez |
| Horizon 2020 Framework Programme | Laserlab-Europe GA no. 871124 | Pablo Loza-Alvarez |
| Fundació Privada Cellex | | Matteo Bernardello<br>Pablo Loza-Alvarez<br>Emilio J Gualda |
| Fundación Mig-Puig | | Matteo Bernardello<br>Emilio J Gualda<br>Pablo Loza-Alvarez |

The funders had no role in study design, data collection, and interpretation, or the decision to submit the work for publication.

## Author contributions

Jade Leiba, Conceptualization, Resources, Data curation, Formal analysis, Validation, Investigation, Visualization, Methodology, Writing – original draft, Writing – review and editing; Tamara Sipka, Christina Begon-Pescia, Data curation, Formal analysis, Validation, Investigation, Writing – review and editing; Matteo Bernardello, Resources, Data curation, Investigation, Writing – review and editing; Sofiane Tairi, Data curation, Formal analysis, Investigation; Lionello Bossi, Resources, Writing – review and editing; Anne-Alicia Gonzalez, Investigation; Xavier Mialhe, Data curation, Formal analysis, Visualization; Emilio J Gualda, Resources, Supervision, Investigation, Writing – review and editing; Pablo Loza-Alvarez, Resources, Supervision, Funding acquisition, Writing – review and editing; Anne Blanc-Potard, Conceptualization, Methodology, Writing – review and editing; Georges Lutfalla, Conceptualization, Funding acquisition, Writing – review and editing; Mai E Nguyen-Chi, Conceptualization, Formal analysis, Supervision, Funding acquisition, Methodology, Writing – original draft, Project administration, Writing – review and editing

## Author ORCIDs

Pablo Loza-Alvarez ⓘ http://orcid.org/0000-0002-3129-1213
Mai E Nguyen-Chi ⓘ http://orcid.org/0000-0003-2672-2426

## Ethics

Animal experimentation procedures were carried out according to the European Union guidelines for handling of laboratory animals (https://ec.europa.eu/environment/chemicals/lab_animals/index_en.htm) and were approved by the Comité d'Ethique pour l'Expérimentation Animale under reference CEEA-LR-B4-172-37 and APAFIS #36309-2022040114222432 V2. Fish husbandry, embryo collection, animal experimentations, handling, and euthanasia were performed at the University of Montpellier, LPHI/CNRS UMR5295, by authorized staff. All experimentations were performed under tricaine (ethyl 3-aminobenzoate) anesthesia, and every effort was made to minimize suffering. Euthanasia was performed using an anesthetic overdose of tricaine.

## Decision letter and Author response

Decision letter https://doi.org/10.7554/eLife.89828.sa1
Author response https://doi.org/10.7554/eLife.89828.sa2

# Additional files

## Supplementary files
- MDAR checklist

## Data availability

The raw sequencing data are available in the NCBI GEO database under accession number: GSE224985, https://www.ncbi.nlm.nih.gov/geo/query/acc.cgi?acc=GSE224985. *Figure 7—source data 1* contains the numerical data used to generate the figures. Other data that support the findings are openly available from the public repository Zenodo at https://zenodo.org/records/10409519.

The following datasets were generated:

| Author(s) | Year | Dataset title | Dataset URL | Database and Identifier |
|---|---|---|---|---|
| Laiba J, Begon-Pescia C, Nguyen-Chi M | 2023 | Dynamic changes of macrophage polarization during *Salmonella* infection in zebrafish | https://www.ncbi.nlm.nih.gov/geo/query/acc.cgi?acc=GSE224985 | NCBI Gene Expression Omnibus, GSE224985 |
| Laiba J, Begon-Pescia C, Sipka T, Tairi S, Nguyen-Chi M | 2023 | Dynamics of macrophage polarization in *Salmonella* infection : Raw data | https://zenodo.org/records/10409519 | Zenodo, 10.5281/zenodo.10409519 |

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
