## [Editor Report]

This useful study introduces the development of *Salmonella* infection model in zebrafish embryos as an important model to study the interaction between macrophages and *Salmonella* during in vivo infection. Overall, the data presented are convincing and provide an inventory of genes mediating macrophage cell-cell adhesion and interactions that are useful for dissecting tissue macrophage responses and heterogeneity during intracellular bacterial infection. This is important to characterise the infection outcome and the dynamics of the immune response. The work will be of interest to microbiologists.

---

## [Decision Letter]

**Decision letter after peer review:**

Thank you for submitting your article "Dynamics of macrophage polarization support *Salmonella* persistence in a whole living organism" for consideration by *eLife*. Your article has been reviewed by 2 peer reviewers, and the evaluation has been overseen by a Reviewing Editor and Bavesh Kana as the Senior Editor. The reviewers have opted to remain anonymous. Please excuse the extraordinary long time it took us to handle the manuscript, due to summer vacations.

Essential revisions:

1) The study proposes that a shift in macrophage polarization from acute to persistent infection stage supports *Salmonella* persistence. Quantitative analyses demonstrating frequencies of infected macrophages and intracellular bacterial levels of infected macrophages during persistent vs. acute infection are not shown. It is also not clear from figure 1E that the total tissue bacterial levels at persistent infection stage (4 dpi) is the same, less, or more compared to 3 dpi, 2 dpi, and 1 dpi, without or without including the infected larvae with high bacterial proliferation. Please provide this.

2) Transcriptional differences in cell adhesion genes of macrophages between acute and persistent infection stage would benefit from validation with representative protein expression analysis. In addition, whether availability of chemotaxis signals and/or cell density contribute to the low motility of macrophages in persistent infection stage, compared to the acute stage, has not been specifically addressed. Please address this.

3) in vivo macrophages are very heterogenous and functionally diverse. The dynamic differences in bulk population transcriptomics of macrophages between the acute and persistent infection stage likely reflect differential recruitment, proliferation, cell death, and polarization that change the relative abundances among different macrophage populations at the site of infection from one time point to another, rather than just a "switch" from M1-like to non-M1 like polarization of the same or an equivalent cell population over time. Having a more robust discussion of these possibilities would not reduce the impact of this study and could help encourage future studies in the field. Please discuss this point.

4) Here authors talk about 'persistent *Salmonella*’: the concept here needs a little care. The authors are more talking about 'chronic infections' than persistent after antibiotic treatment. Can you please specify this carefully, so we avoid another layer of confusion on the topic?

*Reviewer #1 (Recommendations for the authors):*

In this manuscript, the authors seek to investigate the spatiotemporal dynamics of macrophage polarization during *Salmonella* infection. They undertake intravital microscopy of *Salmonella* Typhimurium infection in the hindbrain ventricle of zebrafish larvae and couple this with transcriptomic analysis of macrophages from infected tissues. They find that macrophages and neutrophils are rapidly recruited to the site of infection within hours after inoculation. Macrophage abundance is significantly increased in the persistent infection stage at 4 days post-inoculation (dpi), compared to in the early stage, hours post-inoculation. The authors observe that *Salmonella* bacilli selectively co-localize with aggregates of macrophages, but not neutrophils, during persistent infection. Furthermore, they show that in early infection stage, a markedly higher fraction of macrophages at the site of infection expressed tnfa and exhibits stronger transcriptional signature of pro-inflammatory, M1-like phenotype, compared to macrophages in persistent infection stage. Additionally, the authors find that genes involved in cell-cell adhesion are down-regulated in persistent stage macrophages and these cells have reduced motility. This study's approach, further developing and employing a zebrafish S. Typhimuirum infection model and intravital microscopy of whole living animals, presents an exciting strategy to investigate macrophage responses and their roles during vacuolar intracellular bacterial infection in vivo, complementary to the more commonly utilized murine infection models. The study's findings are useful and largely observational. The data presented have the potential but additional analyses and experiments are needed to clarify and support the conclusions.

Please see below for my specific comments that I hope could be useful for the authors and help strengthen this manuscript:

1) The study proposes that a shift in macrophage polarization from acute to persistent infection stage supports *Salmonella* persistence. Model figure 8 shows M1-like, pro-inflammatory macrophages in acute infection; whereas, in persistent infection there are more macrophages and they are non-M1 like, anti-inflammatory macrophages with higher intracellular bacterial levels. However, quantitative analyses demonstrating frequencies of infected macrophages and intracellular bacterial levels of infected macrophages during persistent vs. acute infection are not shown. It is also not clear from figure 1E that the total tissue bacterial levels at persistent infection stage (4 dpi) is the same, less, or more compared to 3 dpi, 2 dpi, and 1 dpi, without or without including the infected larvae with high bacterial proliferation.

2) I think quantitative analyses for cellular features, motility, spatial distribution, etc. among uninfected bystander macrophages vs. infected macrophages with low intracellular bacterial levels vs. infected with high bacterial levels during acute and persistent infection stage would provide valuable insights.

3) Line 599-600, it appears that only larvae that had macrophage clusters in the brain at 4 dpi were used for transcriptomics analysis. What is the rationale for this inclusion/exclusion and how may this factor into the differences in macrophage transcriptional signatures between the compared samples?

4) It would be useful to note the purity of sorted macrophage populations that underwent transcriptomics analysis.

5) in vivo macrophages are very heterogenous and functionally diverse. The dynamic differences in bulk population transcriptomics of macrophages between the acute and persistent infection stage likely reflect differential recruitment, proliferation, cell death, and polarization that change the relative abundances among different macrophage populations at the site of infection from one time point to another, rather than just a "switch" from M1-like to non-M1 like polarization of the same or an equivalent cell population over time.

6) Transcriptional differences in cell adhesion genes of macrophages between acute and persistent infection stage would need validation with representative protein expression analysis. In addition, whether availability of chemotaxis signals and/or cell density contribute to the low motility of macrophages in persistent infection stage, compared to the acute stage, has not been specifically addressed.

7) Line 410 mentioned that during persistent phase, neutrophils "are poorly immobilized" (data not shown?). Assessing the specificity of cellular motility change from acute to persistent infection would provide useful insights.

*Reviewer #2 (Recommendations for the authors):*

All the best for the revision of the story. The zebrafish model is really great and will advance massively our understanding of infection processes.

---

## [Author Response]

Essential revisions:1) The study proposes that a shift in macrophage polarization from acute to persistent infection stage supports *Salmonella* persistence. Quantitative analyses demonstrating frequencies of infected macrophages and intracellular bacterial levels of infected macrophages during persistent vs. acute infection are not shown. It is also not clear from figure 1E that the total tissue bacterial levels at persistent infection stage (4 dpi) is the same, less, or more compared to 3 dpi, 2 dpi, and 1 dpi, without or without including the infected larvae with high bacterial proliferation. Please provide this.

We agree with the reviewer that quantitative analyses demonstrating frequencies of infected macrophages among recruited macrophage population at acute (4 hpi) and persistent (4 dpi) infection stage were missing and we provide this analysis in the revised manuscript. Between 3 and 14 hpi bacteria can be observed outside leukocytes, but they were actively engulfed by neutrophils and macrophages and 20 to 30% of HBV macrophages were infected. During persistent stages (4 dpi), bacteria were predominantly associated with macrophages, and approximately 60% of the HBV macrophage population was infected. This analysis is now included in the new Figure 5A.

We have also compared intracellular bacterial levels of infected macrophages between these two stages. An accurate quantification of the number of intracellular bacteria per macrophages on microscopy 3D-datasets is challenging because bacteria are aggregated inside the cells. Therefore, we have quantified the size of bacterial aggregates (E2Crimson-positive events) visualized by confocal microscopy. At 5 hpi, the size of E2Crimson-positive aggregates (E2Crimson^+^) was lower than that at 4 dpi. The size distribution analysis of E2Crimson^+^ events indicated a higher representation of smaller sizes (0.5-1.5 µm^3^ and 1.5-10 µm^3^) at 5 hpi compared to 4 dpi, a stage during which very large E2Crimson^+^ events were observed (100-1000 µm^3^, and up to more than 1000 µm^3^). This analysis, which supports an elevated number of bacteria within macrophages during persistent stages, is now included in the new Figure 5B-D. Results and Methods sections have been modified accordingly.

Further, in Figure 1E we carried out a statistical analysis to determine the differences in total bacterial load at different times, including or not the infected larvae with high bacterial proliferation. We observed no significant differences between the group, regardless infected larvae with high bacterial proliferation were included or not. This analysis is now included in the legend of Figure 1.

2) Transcriptional differences in cell adhesion genes of macrophages between acute and persistent infection stage would benefit from validation with representative protein expression analysis. In addition, whether availability of chemotaxis signals and/or cell density contribute to the low motility of macrophages in persistent infection stage, compared to the acute stage, has not been specifically addressed. Please address this.

We agree that gene expression differences in macrophages between acute and persistent infection stage would benefit from further validation with other approaches.

First, we chose to validate the regulation of wt1b, a gene known to be expressed in

regenerative macrophages and required for heart and fin regeneration in zebrafish. In our study, the transcriptomic analysis showed that wt1b mRNA is specifically upregulated in macrophages at 4 dpi but not at 4 hpi. To validate this upregulation, we crossed Tg(wt1b:GFP) line with Tg(mfap4:mCherryF) line to track simultaneously wt1b expressing cells and macrophages. Double transgenic embryos were infected and subsequently imaged at 4hpi and 4 dpi. Confocal microscopy analysis revealed that,

during acute infection (4 hpi) macrophages exhibited no expression of wt1b, while they strongly expressed wt1b at 4 dpi. Persistent infection stage was characterized by the presence of large clusters of wt1b+ macrophages. This confirms that the pro-regenerative gene wt1b is upregulated in macrophages during persistent stage, supporting polarization toward a pro-regenerative state; this also reveals phenotypical distinct macrophage signatures during early and persistent infection stages that can be identified in situ. This data is now included in new Figure 8. Results and Methods sections have been modified accordingly.

Second, we focused on the cell adhesion related gene zo-1 (also known as tjp1a), which was regulated during *Salmonella* infection as shown by RNA sequencing. We studied Zo-1 expression in macrophages at protein level by staining infected Tg(mfap4:mCherryF) embryos with an anti-Zo-1 antibody and an anti-mCherry antibody to visualize macrophages. Immunodetection of Zo-1 protein in zebrafish embryo was confirmed by the labelling of enveloping cell layer (EVL) at 50% epiboly as previously shown (Schwayer et al. 2019 -https://doi.org/10.1016/j.cell.2019.10.006). EVL (keratinocytes) were also labelled at 3 and 6 dpf. We found that Zo-1 was present at the membrane of some amoeboid macrophages at 4 hpi. By contrast, we did not detect Zo-1 labelling in clustered macrophages at 4 dpi which is consistent with our transcriptomic analysis. Because an accurate quantification is prevented by the high background of the ZO-1 staining, we decided not to include this data in the article. The reviewer can visualize the data obtained in “Figure for reviewers”. Further, we have been looking for candidates to validate gene regulation at protein level, but unfortunately, very few antibodies are available for zebrafish, which is not a model commonly used for protein expression studies.

Regarding the availability of chemotaxis signals and/or cell density, we thank the reviewer for this suggestion. Analysis of leukocyte motility revealed that both macrophages and neutrophils were highly mobile during acute infection stage, moving with high speed. By contrast, during persistent infection stage, clustered macrophages (bystanders and infected) had a significant decreased motility compared to control. Importantly, neutrophils did not change their motility during persistent stages compared to control. The fact that only macrophages were motionless, while neutrophils moved normally, suggests that cell density does not influence the motility of leukocytes in persistent infection stage. This data is now included in new Figure 9D and new Figure 9—figure supplement 1. Results and Methods sections have been modified accordingly.

Chemokines play a pivotal role as primary regulators of macrophage chemotaxis, and alterations in these signaling pathways may contribute to the diminished motility of macrophages during persistent infection stages. Previous studies have established the involvement of Ccl2 and its receptor Ccr2 in guiding macrophages towards bacterial pathogens in mice and zebrafish (Cambier et al, 2014). Using RT-qPCR, we showed the upregulation of both ccr2 and ccl2 orthologue, ccl38a.4, following *Salmonella* infection at 4 hpi, yet their expression reverted to baseline levels by 4 dpi. In contrast, *Salmonella* infection had no impact on the expression of cxcr4b and sdf1 (cxcl12a), representing another crucial chemoattractant signal that regulates monocyte-macrophage differentiation (Sánchez-Martín et al. 2011). Furthermore, our transcriptomic analysis revealed that both cxcr3 and cxcl11 are regulated in response to *Salmonella* infection. Notably, the levels of cxcl11.1 were markedly higher at 4 hpi compared to 4 dpi, while cxcl11.5, cxcl11.6, cxcl11.7, cxcr3.1 and cxcr3.2 were all down regulated at 4 hpi. The previously established role of the CXCR3-CXCL11 axis in mediating macrophage recruitment during mycobacterial infections adds significance to these findings(Torraca et al. 2025-DOI: 10.1242/dmm.017756). In summary, our comprehensive dataset suggests a dynamic shift in major chemotaxis signals between the acute and persistent stages of *Salmonella* infection. Therefore, chemokines availability may also contribute to the regulation of macrophage motility during the different phases of bacterial infection. These points have been discussed in the “discussion” section.

3) in vivo macrophages are very heterogenous and functionally diverse. The dynamic differences in bulk population transcriptomics of macrophages between the acute and persistent infection stage likely reflect differential recruitment, proliferation, cell death, and polarization that change the relative abundances among different macrophage populations at the site of infection from one time point to another, rather than just a "switch" from M1-like to non-M1 like polarization of the same or an equivalent cell population over time. Having a more robust discussion of these possibilities would not reduce the impact of this study and could help encourage future studies in the field. Please discuss this point.

We fully agree that this point has to be discussed. We have now included in the Discussion section the following text:

“It is important to highlight that the dynamic differences in macrophage transcriptomics between the acute and persistent infection stages can reflect discrepancies in proliferation, death, and polarization, altering the composition of macrophage populations over time. However, it might also denote a phenotypic switch from M1-like to M2- like polarization within individual macrophages over time. In the future, a more comprehensive analysis at single cell resolution should better discriminate phenotypically distinct macrophage populations at play in this model”.

4) Here authors talk about 'persistent *Salmonella*’: the concept here needs a little care. The authors are more talking about 'chronic infections' than persistent after antibiotic treatment. Can you please specify this carefully, so we avoid another layer of confusion on the topic?

We agree that a clarification of the concept of 'persistent Salmonella' is important. In this study “persistent *Salmonella* infection” refers to an infection wherein a portion of bacteria persists for an extended period within the host. We do not share the reviewers’ view on the fact that this should be “chronic infections”. While our zebrafish model shows Salmonella persistence for 14 days, the majority of experiments have been limited to a duration of 4 days. This limitation prevents us from categorizing our findings as indicative of a chronic infection. The "persistent bacteria" mentioned in the study should not be confused with "Persisters", which refer to bacteria that persist despite antibiotic treatment (Fisher et al. 2017). Persistent bacteria and persisters may share common features, as growth-arrested state, but we have not yet addressed this point. We have now clarified the meaning of "persistent bacteria" in the “Discussion” section.

Reviewer #1 (Recommendations for the authors):Please see below for my specific comments that I hope could be useful for the authors and help strengthen this manuscript:1) The study proposes that a shift in macrophage polarization from acute to persistent infection stage supports *Salmonella* persistence. Model figure 8 shows M1-like, pro-inflammatory macrophages in acute infection; whereas, in persistent infection there are more macrophages and they are non-M1 like, anti-inflammatory macrophages with higher intracellular bacterial levels. However, quantitative analyses demonstrating frequencies of infected macrophages and intracellular bacterial levels of infected macrophages during persistent vs. acute infection are not shown. It is also not clear from figure 1E that the total tissue bacterial levels at persistent infection stage (4 dpi) is the same, less, or more compared to 3 dpi, 2 dpi, and 1 dpi, without or without including the infected larvae with high bacterial proliferation.

We agree with the reviewer that quantitative analyses demonstrating frequencies of infected macrophages among recruited macrophage population at acute (4 hpi) and persistent (4 dpi) infection stage were missing and we provide this analysis in the revised manuscript. Between 3 and 14 hpi bacteria can be observed outside leukocytes, but they were actively engulfed by neutrophils and macrophages and 20 to 30% of HBV macrophages were infected. During persistent stages (4 dpi), bacteria were predominantly associated with macrophages, and approximately 60% of the HBV macrophage population was infected. This analysis is now included in the new Figure 5A.

We have also compared intracellular bacterial levels of infected macrophages between these two stages. An accurate quantification of the number of intracellular bacteria per macrophages on microscopy 3D-datasets is challenging because bacteria are aggregated inside the cells. Therefore, we have quantified the size of bacterial aggregates (E2Crimson-positive events) visualized by confocal microscopy. At 5 hpi, the size of E2Crimson-positive aggregates (E2Crimson+) was lower than that at 4 dpi. The size distribution analysis of E2Crimson+ events indicated a higher representation of smaller sizes (0.5-1.5 µm3 and 1.5-10 µm3) at 5 hpi compared to 4 dpi, a stage during which very large E2Crimson+ events were observed (100-1000 µm3, and up to more than 1000 µm3). This analysis, which supports an elevated number of bacteria within macrophages during persistent stages, is now included in the new Figure 5B-D. Results and Methods sections have been modified accordingly.

Further, in Figure 1E we carried out a statistical analysis to determine the differences in total bacterial load at different times, including or not the infected larvae with high bacterial proliferation. We observed no significant differences between the group, regardless infected larvae with high bacterial proliferation were included or not. This analysis is now included in the legend of Figure 1.

2) I think quantitative analyses for cellular features, motility, spatial distribution, etc. among uninfected bystander macrophages vs. infected macrophages with low intracellular bacterial levels vs. infected with high bacterial levels during acute and persistent infection stage would provide valuable insights.

We fully agree with the reviewers that adding quantitative analyses for motility among

uninfected bystander macrophages vs. infected macrophages during acute and persistent infection stage would enrich the study. Therefore, we have incorporated a new analysis of cell motility to Figure 8, new panel D. Tracking individual infected macrophages in time-lapse video sequences is challenging because macrophages strongly aggregated, preventing to differentiate the cells and their paths. To

accurately quantify cellular features of uninfected bystander macrophages and infected macrophages during infection, we developed a novel transgenic line, Tg(mfap4:Gal4/UAS:nfsb-mCh), in which the mCherry protein is strongly and mosaically expressed in macrophages. We showed distinct dynamics in macrophage behavior during acute and persistent infections. Notably, bystander macrophages

exhibited higher velocity during acute infection, while infected macrophages maintained a speed comparable to the control group. In contrast, during persistent infection, both bystander and infected macrophages exhibited reduced velocity compared to unstimulated macrophages. This observation underscores the dynamic cellular characteristics exhibited by macrophages, influenced by their infection status and the stage of the infection. This data is now included in new figure 9. Results and

Methods sections have been modified accordingly.

We agree that exploring other cellular features and spatial distribution would be highly interesting. A precise analysis of the structure of cellular aggregates and the characteristics of the cells involved is a point that deserves further exploration.

3) Line 599-600, it appears that only larvae that had macrophage clusters in the brain at 4 dpi were used for transcriptomics analysis. What is the rationale for this inclusion/exclusion and how may this factor into the differences in macrophage transcriptional signatures between the compared samples?

We acknowledge that further explanation is needed. As shown in Figure 1, *Salmonella* induces different outcomes in zebrafish larvae at late time points (4 dpi), i.e. "high proliferation", "infected" or "cleared". This may result in different transcriptomic profiles of macrophages. We thus anticipated that bulk RNA sequencing would not provide meaningful biological signals and would be difficult to interpret. Therefore, we decided to focuse on the “infected” cohort that displays persistent infection at 4 dpi, and harbors macrophage clusters in the brain. The rational for inclusion/exclusion of larvae is now provided in the method section.

4) It would be useful to note the purity of sorted macrophage populations that underwent transcriptomics analysis.

We agree that this information is important. Preliminary experiments were performed to set up FACS sorting procedure from dissociated Tg(mfap4:mcherry-F) larvae and RNA preparation (Begon-Pescia et al. 2022). We validated the identity of sorted zebrafish macrophage populations by monitoring mfap4 mRNA expression using RT-qPCR on mCherry positive and mCherry negative sorted cells. The results showed that mCherry positive cells expressed 1000 time more mfap4 mRNA than mCherry negative cells, demonstrating that FACS sorting protocol allowed enrichment in macrophage populations. This data is now included in new Figure 7—figure supplement 1. To further validate that the obtained transcriptomic profiles shared common features and corresponded to macrophage transcriptomic profiles, we checked the number of normalized reads corresponding to cell specific markers. We noticed that all samples expressed considerable levels of several key myeloid makers, namely, mfap4, mpeg1.2, marco, csf1ra and lygl1 but they did not express markers for other cell types (mucus secreting cells, xantophores, pronephros, endoderm, skeletal muscle, epidermis, peridermis, notochord, red blood cells, neutrophils and fibroblast). These data are consistent with the common macrophage features shared by sequenced samples and confirm the purity of sorted macrophage populations. It is now included in the “results” section and in Figure 7 and figure 7—figure supplement 3.

5) in vivo macrophages are very heterogenous and functionally diverse. The dynamic differences in bulk population transcriptomics of macrophages between the acute and persistent infection stage likely reflect differential recruitment, proliferation, cell death, and polarization that change the relative abundances among different macrophage populations at the site of infection from one time point to another, rather than just a "switch" from M1-like to non-M1 like polarization of the same or an equivalent cell population over time.

We fully agree that this point has to be discussed. We have now included in the Discussion section the following text:

“It is important to highlight that the dynamic differences in macrophage transcriptomics between the acute and persistent infection stages can reflect discrepancies in proliferation, death, and polarization, altering the composition of macrophage populations over time. However, it might also denote a phenotypic switch from M1-like to M2- like polarization within individual macrophages over time. In the future, a more comprehensive analysis at single cell resolution should better discriminate phenotypically distinct macrophage populations at play in this model”.

6) Transcriptional differences in cell adhesion genes of macrophages between acute and persistent infection stage would need validation with representative protein expression analysis.

We agree that gene expression differences in macrophages between acute and persistent infection stage would benefit from further validation with other approaches.

First, we chose to validate the regulation of wt1b, a gene known to be expressed in

regenerative macrophages and required for heart and fin regeneration in zebrafish. In our study, the transcriptomic analysis showed that wt1b mRNA is specifically upregulated in macrophages at 4 dpi but not at 4 hpi. To validate this upregulation, we crossed Tg(wt1b:GFP) line with Tg(mfap4:mCherryF) line to track simultaneously wt1b expressing cells and macrophages. Double transgenic embryos were infected and subsequently imaged at 4hpi and 4 dpi. Confocal microscopy analysis revealed that,

during acute infection (4 hpi) macrophages exhibited no expression of wt1b, while they strongly expressed wt1b at 4 dpi. Persistent infection stage was characterized by the presence of large clusters of wt1b+ macrophages. This confirms that the pro-regenerative gene wt1b is upregulated in macrophages during persistent stage, supporting polarization toward a pro-regenerative state; this also reveals phenotypical distinct macrophage signatures during early and persistent infection stages that can be identified in situ. This data is now included in new Figure 8. Results and Methods sections have been modified accordingly.

Second, we focused on the cell adhesion related gene zo-1 (also known as tjp1a), which was regulated during *Salmonella* infection as shown by RNA sequencing. We studied Zo-1 expression in macrophages at protein level by staining infected Tg(mfap4:mCherryF) embryos with an anti-Zo-1 antibody and an anti-mCherry antibody to visualize macrophages. Immunodetection of Zo-1 protein in zebrafish embryo was confirmed by the labelling of enveloping cell layer (EVL) at 50% epiboly as previously shown (Schwayer et al. 2019 -https://doi.org/10.1016/j.cell.2019.10.006). EVL (keratinocytes) were also labelled at 3 and 6 dpf. We found that Zo-1 was present at the membrane of some amoeboid macrophages at 4 hpi. By contrast, we did not detect Zo-1 labelling in clustered macrophages at 4 dpi which is consistent with our transcriptomic analysis. Because an accurate quantification is prevented by the high background of the ZO-1 staining, we decided not to include this data in the article. The reviewer can visualize the data obtained in “Figure for reviewers”. Further, we have been looking for candidates to validate gene regulation at protein level, but unfortunately, very

few antibodies are available for zebrafish, which is not a model commonly used for protein expression studies.

In addition, whether availability of chemotaxis signals and/or cell density contribute to the low motility of macrophages in persistent infection stage, compared to the acute stage, has not been specifically addressed.

We thank the reviewer for this suggestion.

Analysis of leukocyte motility revealed that both macrophages and neutrophils were highly mobile during acute infection stage, moving with high speed. By contrast, during persistent infection stage, clustered macrophages (bystanders and infected) had a significant decreased motility compared to control. Importantly, neutrophils did not change their motility during persistent stages compared to control. The fact that only macrophages were motionless, while neutrophils moved normally, suggests that cell density does not influence the motility of leukocytes in persistent infection stage. This data is now included in new Figure 9D and new Figure 9—figure supplement 1. Results and Methods sections have been modified accordingly.

Chemokines play a pivotal role as primary regulators of macrophage chemotaxis, and

alterations in these signaling pathways may contribute to the diminished motility of macrophages during persistent infection stages. Previous studies have established the involvement of Ccl2 and its receptor Ccr2 in guiding macrophages towards bacterial pathogens in mice and zebrafish (Cambier et al, 2014). Using RT-qPCR, we showed the upregulation of both ccr2 and ccl2 orthologue, ccl38a.4, following Salmonella infection at 4 hpi, yet their expression reverted to baseline levels by 4 dpi. In contrast, Salmonella infection had no impact on the expression of cxcr4b and sdf1 (cxcl12a), representing another crucial chemoattractant signal that regulates monocyte-macrophage

differentiation (Sánchez-Martín et al. 2011). Furthermore, our transcriptomic analysis revealed that both cxcr3 and cxcl11 are regulated in response to Salmonella infection. Notably, the levels of cxcl11.1 were markedly higher at 4 hpi compared to 4 dpi, while cxcl11.5, cxcl11.6, cxcl11.7, cxcr3.1 and cxcr3.2 were all down regulated at 4 hpi. The previously established role of the CXCR3-CXCL11 axis in mediating macrophage recruitment during mycobacterial infections adds significance to these findings

(Torraca et al. 2025-DOI: 10.1242/dmm.017756). In summary, our comprehensive dataset suggests a dynamic shift in major chemotaxis signals between the acute and persistent stages of *Salmonella* infection. Therefore, chemokines availability may also contribute to the regulation of macrophage motility during the different phases of bacterial infection. These points have been discussed in the “discussion” section.

7) Line 410 mentioned that during persistent phase, neutrophils "are poorly immobilized" (data not shown?). Assessing the specificity of cellular motility change from acute to persistent infection would provide useful insights.

We apology for the misunderstanding. We just mentioned that during persistent phase, neutrophils are poorly recruited. We modified the text to avoid any ambiguity. However, we do agree that neutrophil motility merits further exploration and we have monitored neutrophil motility and velocity and have incorporated this analysis to Figure 9—figure supplement 1.